

# Specific carotenoid pigments in the diet and a bit of oxidative stress in the recipe for producing red carotenoid-based signals

Esther García-de Blas[1], Rafael Mateo[1] and Carlos Alonso-Alvarez[1,2]

[1] Instituto de Investigación en Recursos Cinegéticos (IREC), CSIC-UCLM-JCCM, Ciudad Real, Spain
[2] Ecología Evolituva, Museo Nacional de Ciencias Naturales (MNCN), Consejo Superior de Investigaciones Científicas (CSIC), Madrid, Spain

Corresponding author
Carlos Alonso-Alvarez,
carlos.alonso@csic.es

## ABSTRACT

Colorful ornaments have been the focus of sexual selection studies since the work of Darwin. Yellow to red coloration is often produced by carotenoid pigments. Different hypotheses have been formulated to explain the evolution of these traits as signals of individual quality. Many of these hypotheses involve the existence of a signal production cost. The carotenoids necessary for signaling can only be obtained from food. In this line, carotenoid-based signals could reveal an individual's capacity to find sufficient dietary pigments. However, the ingested carotenoids are often yellow and became transformed by the organism to produce pigments of more intense color (red ketocarotenoids). Biotransformation should involve oxidation reactions, although the exact mechanism is poorly known. We tested the hypothesis that carotenoid biotransformation could be costly because a certain level of oxidative stress is required to correctly perform the conversion. The carotenoid-based signals could thus reveal the efficiency of the owner in successfully managing this challenge. In a bird with ketocarotenoid-based ornaments (the red-legged partridge; *Alectoris rufa*), the availability of different carotenoids in the diet (i.e. astaxanthin, zeaxanthin and lutein) and oxidative stress were manipulated. The carotenoid composition was analyzed and quantified in the ornaments, blood, liver and fat. A number of oxidative stress biomarkers were also measured in the same tissues. First, we found that color and pigment levels in the ornaments depended on food levels of those carotenoids used as substrates in biotransformation. Second, we found that birds exposed to mild levels of a free radical generator (diquat) developed redder bills and deposited higher amounts of ketocarotenoids (astaxanthin) in ornaments. Moreover, the same diquat-exposed birds also showed a weaker resistance to hemolysis when their erythrocytes were exposed to free radicals, with females also enduring higher oxidative damage in plasma lipids. Thus, higher color production would be linked to higher oxidative stress, supporting the biotransformation hypothesis. The recent discovery of an avian oxygenase enzyme involved in converting yellow to red carotenoids may support our results. Nonetheless, the effect could also depend on the abundance of specific substrate carotenoids in the diet. Birds fed with proportionally higher levels of zeaxanthin showed the reddest ornaments with the highest astaxanthin concentrations. Moreover, these birds tended to show the strongest diquat-mediated effect. Therefore, in the evolution of carotenoid-based sexual signals, a biotransformation

cost derived from maintaining a well-adjusted redox machinery could coexist with a cost linked to carotenoid acquisition and allocation (i.e. a resource allocation trade-off).

## INTRODUCTION

Colored ornaments in animals have attracted the attention of evolutionary biologists since Charles Darwin, who suggested that most conspicuously colored traits are the product of sexual selection (*Darwin, 1871*). Colored ornaments should provide some advantage when competing for a mate with same sex individuals (intrasexual selection) or by being more attractive to the choosing sex (intersexual selection; *Andersson, 1994*). In many cases, colored traits inform competitors or potential mates about the quality of the owner. However, the trait should generate some benefit for both emitter and receptor to be considered as a signal (*Hasson, 1997*; *Bradbury & Vehrenkamp, 1998*; *Maynard Smith & Harper, 2003*). This can occur by the transmission of information in a reliable (non-falsifiable) way (*Maynard Smith & Harper, 2003*).

*Zahavi (1975)* proposed the "handicap principle," in which the reliability of the signal is due to its production/maintenance costs. The expression of a signal would proportionally be more costly for low-quality individuals compared to high-quality ones (*Grafen, 1990*; also *Getty, 2006*), the former being unable to signal or signaling in an inefficient way.

Carotenoids are natural pigments with immune-stimulant and antioxidant properties (*Britton, Liaaen-Jensen & Pfander, 2009*) that are present in the integument of many vertebrate species, generating conspicuously colored traits (e.g. *Brush, 1990*; *Stradi, 1998*; *McGraw, 2006*). The most obvious cost of carotenoid-based signals is the increase of conspicuousness that would raise the risk of predation (e.g. *Godin & McDonough, 2003*). This idea was suggested as early as *Darwin (1871)*, regarding colorful ornaments but without citing the pigments.

The second cost associated with these traits is related to the fact that carotenoids cannot be synthesized de novo by the organism, but are only obtained from food (*Britton, Liaaen-Jensen & Pfander, 2009*; *McGraw, 2006*). Assuming that carotenoids are relatively scarce in food, colored individuals should pay a cost in terms of energy or time spent searching for pigments, which was suggested by *Endler (1980)* and *Endler (1983)* in fish studies (also *Kodric-Brown, 1985*; see in birds *Hill, 1990*; *McGraw, 2006*). This hypothesis is difficult to test and has garnered mixed support, at least in avian species (reviewed in *Hill (2006)*), which is probably the taxon where carotenoid-based signaling has been studied most in-depth (*McGraw, 2006*; *Pérez-Rodríguez, 2009*; *Simons, Cohen & Verhulst, 2012*). Subsequently, *Lozano (1994)* was the first to emphasize the physiologically specific roles of carotenoids in an evolutionary context, suggesting that investing large

amounts of pigment in signaling could compromise the immune system. This idea seems to be well supported at least for some inflammatory responses (phytohaemagglutinin skin test) in birds (reviewed in *Simons, Cohen & Verhulst (2012)*). Subsequently, *von Schantz et al. (1999)* followed a similar reasoning but regarding the antioxidant properties of the pigments, proposing that investing in coloration would challenge the individual's capacity to combat oxidative stress. This type of stress is the result of an imbalance between the production of reactive oxygen and nitrogen species (RONS) by cell respiration and immune responses and the state (levels and efficiency) of the antioxidant defenses (*Halliwell & Gutteridge, 2007*). An evolutionary trade-off (*van Noordwijk & de Jong, 1986*) in the investment of the carotenoid resources between self-maintenance (antioxidant defense) and reproduction (sexual signaling) could thus be established (*Møller et al., 2000*; *Alonso-Alvarez et al., 2008*). The *von Schantz et al. (1999)* hypothesis has gained popularity (e.g. *Blount et al., 2003*; *Alonso-Alvarez et al., 2004*; *Hörak et al., 2007*), probably because it unifies the physiological components of trait expression, since the immune response is at least partially regulated by the oxidative machinery (*Halliwell & Gutteridge, 2007*; *Sorci & Faivre, 2009*; *Vallverdú-Coll et al., 2015*).

Nonetheless, the antioxidant role of those carotenoids involved in sexual signaling has been questioned. This criticism has mostly arisen from the weakness of some correlations between carotenoid blood levels and certain measures of antioxidant capacity or oxidative damage in avian species (*Costantini & Møller, 2008*; *Isaksson & Andersson, 2008*). However, a meta-analysis on the published literature of this taxon seems to support the carotenoid antioxidant function, although the results were not robust (*Simons, Cohen & Verhulst, 2012*).

Importantly, the carotenoid molecules giving color to the ornaments are frequently not the same as those carotenoids obtained from the diet and circulating in the blood (e.g. fishes: *Hata & Hata (1972)* and *Ohkubo et al. (1999)*; birds: *McGraw (2006)* and references therein). This issue may be key to understanding the cost of the signal, but many obscure points are as yet not understood. In particular, the site (tissue) where carotenoids are transformed and the type of biochemical processes involved in such transformations are little understood.

In avian species, the liver was the first tissue proposed as a potential biotransformation site (*Brush & Power, 1976*; *Brush, 1990*) because it stores large amounts of carotenoids and it is the main 'laboratory' of the organism (*Blem, 2000*; *Britton, Liaaen-Jensen & Pfander, 2009*). Carotenoid biotransformation in the liver could compete with the activity of enzymes involved in detoxification (*Blem, 2000*; *Hill & Johnson, 2012*). Hence, the fact that this vital organ could be involved could affect our understanding of the costs derived from color production. Carotenoid transformation in the liver was supported by studies in crossbills (*Loxia curvirostra*), which found the pigment used for coloration in the liver and blood (*del Val et al., 2009a*; *del Val et al., 2009b*; see also *Hill & Johnson, 2012*). Studies in many other bird species, however, did not find this and instead suggested that the ornament is the main transforming site (*McGraw, 2004*; *McGraw, 2009*; *García-de Blas et al., 2014*), which would perhaps be less important for survival compared to the liver.

To understand how carotenoids are transformed we first need to know the biochemical route followed from substrate pigments to ornamental carotenoids, including the intermediate compounds (*McGraw, 2006*; *Britton, Liaaen-Jensen & Pfander, 2009*). Lutein and zeaxanthin are the most abundant carotenoids in the diet and blood of birds (*McGraw, 2006*). Red ornaments displayed by many animal species are often the result of biotransformation of the cited yellow hydroxycarotenoids in red ketocarotenoids such as astaxanthin or canthaxanthin (*McGraw, 2006*). The pathway followed from hydroxy- to ketocarotenoids requires hydrogenation and oxidation reactions. The existence in vertebrates of specific enzymes (hydroxylases and 4-oxygenases (i.e. ketolases)) was first proposed (*McGraw, 2006*; *Hill & Johnson, 2012*) and subsequently demonstrated in birds (see *Lopes et al. (2016)* and *Mundy et al. (2016)* describing a candidate oxygenase). In this regard, *Hill & Johnson (2012)* and *Johnson & Hill (2013)* have recently suggested that the oxidative status of the organism could influence the activity of these enzymes, with the carotenoid-based signals, in some way, revealing the individual's capacity to efficiently manage oxidative stress. The basic content of this idea was earlier formulated by *Völker (1957)* when trying to explain why wild birds often lost their color in captivity. He proposed that this phenomenon is the result of impairment in the oxidative metabolism involved in carotenoid transformations. Although this could have deep implications for understanding the proximate costs of animal signaling, the hypothesis has not been experimentally tested until now.

In the present study, the red-legged partridge (*Alectoris rufa*) was used as the model species. This gallinacean shows red ornaments (bill, eye rings, and legs) mostly produced by astaxanthin and papilioerythrinone ketocarotenoids (*García-de Blas et al., 2013*; *García-de Blas et al., 2014*). We have experimentally shown that red head traits of males are used by females to adjust their reproductive investment, suggesting that these ornaments are indeed involved in sexual selection (*Alonso-Alvarez et al., 2012*). Experiments have also shown a relationship between integumentary coloration (and circulating carotenoid levels) and individual quality in terms of immune capacity (*Pérez-Rodríguez & Viñuela, 2008*; *Perez-Rodriguez et al., 2008*; *Mougeot et al., 2009*). Redder birds also show a better resistance to oxidative stress when exposed to an immune challenge (*Pérez-Rodríguez, Mougeot & Alonso-Alvarez, 2010*). Moreover, young partridges exposed to high oxidative stress produced paler red traits and circulated lower blood carotenoid levels in adulthood (*Alonso-Alvarez & Galván, 2011*). We have also described that astaxanthin and papilioerythrinone pigments are not present in blood, liver or fat, which indicates that pigment transformation takes place at the ornament site (*García-de Blas et al., 2013*; *García-de Blas et al., 2014*; *García-de Blas, Mateo & Alonso-Alvarez, 2015*). We have proposed that astaxanthin and papilioerythrinone should be derived from zeaxanthin and lutein in food, respectively (i.e. *García-de Blas et al., 2014*), on the basis of published biochemical pathways (*McGraw, 2006*; *LaFountain, Frank & Prum, 2013*). Lutein and zeaxanthin, in this order, are the most abundant carotenoids in the blood of this (*García-de Blas et al., 2013*) and many other bird species (*McGraw, 2006*). As previously noted, the biotransformation of these

compounds should involve oxidative reactions (*McGraw, 2006*). Dietary lutein would be transformed to papilioerythrinone after one 4-oxidation and one dehydrogenation reactions, whereas dietary zeaxanthin would be converted into astaxanthin by two 4-oxidations (*McGraw, 2006*; *LaFountain, Frank & Prum, 2013*; *García-de Blas et al., 2014*).

Here, the carotenoid content of the diet of captive red-legged partridges was manipulated, subsequently exposing birds to an oxidative challenge. Our aims were (1) to reveal the metabolic pathway from dietary carotenoids to those deposited in the ornaments, (2) to verify the contribution to integument coloration of each dietary carotenoid, and (3) to determine if oxidative stress can influence color and the individual capacity to transform substrate carotenoids into those carotenoids allocated to ornaments. In this order, some birds received food supplemented with different zeaxanthin vs. lutein proportions, whereas other individuals received astaxanthin. In order to induce a higher oxidative stress, half of the birds in each treatment were also exposed to a free radical generator (diquat) in drinking water (*Galvan & Alonso-Alvarez, 2009*; see also *Koch & Hill, in press*). We first predicted that a higher proportion of zeaxanthin in the diet should increase astaxanthin levels in ornaments whereas a higher proportion of lutein should instead raise the papilioerythrinone concentration. Since astaxanthin is the most abundant pigment in ornaments (*García-de Blas et al., 2013*; *García-de Blas et al., 2014*), the group receiving dietary astaxanthin should a priori produce the reddest color and the highest astaxanthin concentrations in bare parts because no transformations would be required (*Negro & Garrido-Fernández, 2000*). If transformations depend on specific enzymes inducing oxidative reactions, we can first predict that the oxidative challenge (higher availability of free radicals) could inhibit them by impairing/destabilizing the enzyme such as in the case of well-known antioxidant enzymes whose activity is decreased by high oxidative stress (e.g. glutathione synthase; *Halliwell & Gutteridge, 2007*). This would lead to paler birds with lower ketocarotenoid levels in ornaments. Alternatively, if the oxidative challenge is mild, these reactions could be favored in a sort of compensatory (hormetic) response (e.g. *Costantini, Metcalfe & Monaghan, 2010*). This would lead to redder colors and higher ketocarotenoid levels in ornaments and specific transforming sites (i.e. liver; see above). We can only speculate on this mechanism as the nature of the ketolase enzymes is still poorly understood. Their activity should a priori depend on oxygen availability (*Fraser, Miura & Misawa, 1997*; *Schoefs et al., 2001*). Superoxide or hydrogen peroxide generated by diquat redox cycling (*Fussell et al., 2011*; *Koch & Hill, in press*) could perhaps provide this oxygen required for oxygenase activity and/or activate redox signaling pathways increasing enzyme transcription. In fact, superoxide and hydrogen peroxide can act as prime redox signaling molecules activating many different cell pathways (e.g. *Hurd & Murphy, 2009*). Nonetheless, free radicals derived from diquat redox cycling could also directly promote oxidation of dietary pigments. This last possibility should, however, imply increased ketocarotenoid levels in any body site where pigments and diquat-derived molecules interact (the blood should be the first site after absorption).

## MATERIAL AND METHODS

### Manipulation of carotenoid content in food

In order to manipulate the carotenoid content of the diet, we collaborated with a company dedicated to producing animal pelleted feed (INALSA; Ciudad Real, Spain; http://www.piensos-inalsa.com/contenido/perdices.htm; INALSA-UCLM agreement signed on May 25, 2012). We preferred to manipulate carotenoid levels in food because carotenoids diluted in drinking water (1) can directly pigment head traits due to splashing (previous observations in this and other species) and (2) would have interfered with our oxidative stress manipulation. We supplied a free radical generator (diquat; see below) in water. Carotenoids and diquat in the same solution would have reacted producing pro-oxidant carotenoid metabolites (e.g. *El-Agamey & McGarvey, 2008*). Alternatively, the use of two different water dispensers for each type of treatment would not have guaranteed a similar consumption of each solution.

The manipulation of carotenoid levels in the pellets was made on a basal commercial diet normally used during reproduction of captive red-legged partridges, containing wheat, barley, corn and soy in different proportions (INALSA, Zaragoza, Spain). This feed did not contain any additional carotenoid to those naturally present in the grain (*Panfili, Fratianni & Irano, 2004*) and it was mixed with the different commercial carotenoids resulting in the final feed. Commercial pigments used to prepare the different diets for the experiment were CROMO ORO Classic (min. lutein 16 g/Kg and min. zeaxanthin 0.90 g/Kg), provided by DISPROQUIMA (Barcelona, Spain), OPTISHARP™ (Zeaxanthin 5% CWS/S-TG), provided by DSM Nutritional Products (Switzerland) and CAROPHYLL® Pink (Astaxanthin 10% CWS), provided by DSM Nutritional Products (Madrid, Spain). The adequate amounts of each pigment to add to the food were calculated taking into account the quantities of total carotenoids authorized for poultry feed (Directive 70/524/EEC, Communication 2004/C 50/01). Manipulation of carotenoid levels in the diet should resemble natural scenarios (*Koch, Wilson & Hill, 2016*). However, the natural carotenoid content in the diet of wild red-legged partridges is currently unknown. We should, nonetheless, consider that body carotenoid levels of wild partridges are significantly higher than levels in captive birds that usually receive carotenoid supplements (Table 9S in *García-de Blas, Mateo & Alonso-Alvarez (2015)*). This suggests that our supplements would not produce unnatural phenotypes. Moreover, no negative effect due to a hypothetical pharmacological level was detected in terms of survival, body mass, reproductive output (egg production) or oxidative stress levels (Results).

Pellets were elaborated following the habitual method of commercial feed preparation by using large-scale mills (*Pietsch, 2005*). This process yielded perfectly homogeneous pellets, similar in size and color to base feed, avoiding the pigmentation of the head of the birds by direct contact. Diet 1 (Control) was the basal diet. Diets 2 and 3 contained lutein and zeaxanthin in different proportions: Diet 2 (called LutZea) contained approximately 73% lutein and 27% zeaxanthin, and diet 3 (ZeaLut) was formed by 52% lutein and 48% zeaxanthin. Thus, diet 2 represented proportions often

found in the natural diet of granivorous birds (*McGraw, 2006*), whereas diet 3 was a diet enriched for zeaxanthin. Diet 4 was supplemented with astaxanthin (Ast). Carotenoid, tocopherol and retinol content of each type of pellet are shown in Table 1. Unexpected differences in tocopherol and retinol levels among treatments were found. This was probably due to the protective antioxidant action of carotenoids on vitamins present in the basal feed during the pelleting process, which involves high pressures and temperatures (*Pietsch, 2005*), and to differences in the composition of supplements not detected during the formulation of each diet. Retinol and tocopherol are antioxidant vitamins involved in mutual recycling processes with carotenoids (*Mortensen, Skibsted & Truscott, 2001*; *Catoni, Peters & Schaefer, 2008*; *Surai, 2012*). To discard the influence of this potential bias, tocopherol and retinol levels in every analyzed tissue (ornaments, plasma, liver and fat) were quantified and included as covariates in all statistical models (below).

## Experimental procedure

The study was carried out at the Dehesa de Galiana experimental facilities (Instituto de Investigación en Recursos Cinegéticos and Diputación Provincial, Ciudad Real, Spain). The protocol was approved by the University of Castilla-La Mancha's Committee on Ethics and Animal Experimentation (approval number 1011.01). It was conducted on captive-born, one-year-old red-legged partridges provided by a governmental breeding facility (Chinchilla, Albacete, Spain). We used 182 adult partridges forming 91 pairs that were kept in outdoor cages (1 × 0.5 × 0.4 m, each pair) under natural photoperiods and temperatures. No birds died during the study, but ten birds were removed from the experiment (and statistical analyses) due to escapes during handling (treatment groups did not differ in these exclusions, all $\chi^2$, $P > 0.12$). In these cases, replacement birds were incorporated to keep pairs in similar conditions, but the new birds were not included in posterior samplings. The sex of individuals was determined genetically following *Griffiths et al. (1998)*. Pairs were randomly divided into four groups that received one of the four diets. The sample size for Control, LutZea and ZeaLut groups was 23 pairs, and 22 pairs for the Ast group. Possible differences between groups in terms of food intake were checked during the experiment by weighing the pellet mass in feeders of a subsample of 10 pairs per group during one week, with no difference being detected (repeated-measures ANOVA; $F_{3,80} = 0.732$, $P = 0.536$). The experiment was carried out during the reproductive period (April–June), when the color expression of integuments is the greatest (*Pérez-Rodríguez, 2008*).

On April 11 ("day 0"), a blood sample and a color measurement (below) of each ornament (eye ring, bill, and legs) from each partridge were taken in order to determine pre-treatment color and blood levels of pigments and other physiological variables (below). Color measurements and blood samples were again taken on May 29 ("day 48;" intermediate sample). A third color and blood sampling was performed at the end of the experiment (July 2; "day 82"). One mL of blood was taken from the jugular vein, each time using heparinized syringes. Blood was centrifuged at 10,000 × g for 10 min at 4 °C to separate plasma from the cell fraction. Both were stored separately at

**Table 1 Composition of a sample of each different type of food used in the experiment.** For the names of the diets (see section Manipulation of carotenoid content in food).

| Name diet | Lutein | Zeaxanthin | Astaxanthin | Total carotenoids | Retinol | Tocopherol |
|---|---|---|---|---|---|---|
| Control | 1.33 | 0.69 | 0 | 1.96 | 2.5 | 8.3 |
| LutZea | 24.77 | 9.32 | 0 | 34.09 | 3.5 | 10.9 |
| ZeaLut | 17.64 | 18.6 | 0 | 36.24 | 10.4 | 15.1 |
| Ast | 5.3 | 4.8 | 22.87 | 32.97 | 14.4 | 18.5 |

−80 °C for later analysis. Before centrifugation, an aliquot of each blood sample was taken to calculate the hematocrit and resistance of erythrocytes against an oxidative challenge (see below).

On May 30, just after the second sampling, half of each treatment group ($n = 45$ pairs) were randomly allocated to the oxidative challenge. Of them, 11 pairs were from Control, ZeaLut, and Ast groups, and 12 pairs from the LutZea treatment. These birds were treated with diquat dibromide added to drinking water. The commercial product "Reglone" (Syngenta, Madrid) was used (20% w/v of diquat dibromide in water). Diquat dibromide is a redox cycler that is transformed to a free radical which, in reaction with molecular oxygen, produces superoxide and other redox products (e.g. *Sewalk, Brewer & Hoffman, 2000*; *Zeman et al., 2005*; *Xu et al., 2007*). The diquat bromide dose (i.e. 0.50 mL/L Reglone in drinking water; Reglone contains 20% w/v of diquat dibromide in water) was established on the basis of a pilot study and the results obtained in previous work in the same species, which reported no body mass changes but increased lipid oxidative damage in erythrocytes (see Supporting Fig. 1 in *Alonso-Alvarez & Galván (2011)*; see also *Galvan & Alonso-Alvarez, 2009*).

## Color measurements

The coloration of eye-rings and bills of red-legged partridges was assessed by using a portable spectrophotometer (Minolta CM-2600D, Tokyo). Hue values were calculated by using the formula of *Saks, Mcgraw & Hõrak (2003)* for brightness (B) of different colors (i.e. hue = arctan {[(By−Bb)/BT]/[(Br−Bg)/BT]}, where yellow (y) is the addition of percentage reflectance within the 550–625 nm range, red (r) = 625–700 nm, blue (b) = 400–475 nm, green (g) = 475–550 nm and T is total brightness). BT obtained from our spectrophotometer (360–700 nm) was added as a covariate to models testing the hue (see Statistical Analyses), since the original formula includes BT in both numerator and denominator, thus canceling out its effect. Repeatabilities of triplicate spectrophotometric measurements were significant for both traits ($r > 0.68$, $P < 0.001$), with mean values for each sample being used.

Leg color was assessed by means of digital photographs (Nikon D-3100; see also *García-de Blas et al., 2013*) because the probe of our spectrophotometer did not adapt well to the leg surface (also *Alonso-Alvarez & Galván, 2011*). In this case, the birds were placed in the same position under standardized indoor light conditions (Kaiser Repro Lighting Unit; Repro Base with lights RB260 2 × 11 W 6,000 °K; Kaiser Fototechnik, Buchen)

with the camera (Nikon D-3100) always set to the same focus and conditions. A red color chip (Kodak, NY, USA) was placed close to the legs in order to control for subtle changes in environmental light, adding the hue values of the chip as a covariate to models testing leg color (*Statistical analyses*). Pictures were analyzed by a technician blinded to the birds' identity. The color intensity of the central area of one of the tarsi was determined in adults by recording mean red, green and blue values (RGB system; e.g. *Alonso-Alvarez et al., 2008*) using Adobe Photoshop CS3. Hue was determined after conversion of RGB values by using the *Foley & Van Dam (1984)* algorithm. Repeatability of picture measurements taken twice from a different sample of red-legged partridges was high ($r > 0.90$, $P < 0.001$, $n = 71$; *Alonso-Alvarez & Galván, 2011*). Since lower hue values obtained from spectrophotometer measures or pictures indicated higher redness, the sign of the hue variables was reversed (multiplied by −1) to simplify interpretations. The term "redness" was thus used to describe the hue inverse.

## Quantification of carotenoids and vitamins

The analyses of carotenoids, and vitamins A and E in internal tissues (i.e. plasma, liver, and subcutaneous fat) and colored integuments were performed by HPLC-DAD-FLD following the methods described by *Rodríguez-Estival et al. (2010)*; *García-de Blas et al. (2011)* and *García-de Blas et al. (2013)*. Carotenoid levels are total values adding the levels of esterified and free forms for each specific pigment. Standards of lutein, zeaxanthin, canthaxanthin, astaxanthin, astaxanthin monopalmitate and astaxanthin dipalmitate were purchased from CaroteNature (Lupsingen, Switzerland). Retinyl acetate (used as an internal standard) and standards of retinol and α-tocopherol were provided by Sigma-Aldrich. Carotenoid and vitamin concentrations were expressed as nmoles per gram of tissue.

## Resistance to hemolysis under free radical exposure

The resistance of red blood cells to hemolysis under exposure to a free radical generator was assessed. Whole blood was exposed to a thermo-controlled free radical aggression by adding 2,2-azobis-(aminodinopropane) hydrochloride (AAPH) (*Rojas Wahl et al., 1998*). Previous work has shown that if at least one component of the antiradical detoxification system is impaired, the hemolysis curve shows a shift towards shorter times (*Blache & Prost, 1992*; *Girard et al., 2005*). This test, therefore, provides an assessment of resistance to oxidative stress because all families of free radical scavengers present in the blood are mobilized to fight off the oxidant attack (e.g. *Blache & Prost, 1992*; *Lesgards et al., 2002*; *Girard et al., 2005*). Ten microliters of the blood of adult birds were immediately diluted and mixed with 365 μL of KRL buffer (for 50 mL: 0.020 g of $KHCO_3$; 0.0147 g of $CaCl_2$ $2H_2O$; 0.084 g of $NaHCO_3$; 0.4036 g of NaCl, 0.00746 g of KCl in 50 mL mili-Q water, adjusting pH to 7.4 with 3N HCl). The analyses were performed within 24 h following blood collection. Nonetheless, some aliquots could not be analyzed due to conservation problems, but this did not unbalance sample sizes of CAR and diquat treatments (all $\chi^2$ tests: $P > 0.10$). Eighty microliters of KRL-diluted blood were incubated at 40 °C with 136 μL of a 150 mM solution of AAPH. The lysis of red

blood cells was assessed with a microplate reader device (PowerWave XS2, Bio-Tek Instruments Inc., Winooski, VT), which measures the decrease in optical density at the wavelength of 540 nm every few minutes. Blood samples of a different bird species (zebra finch, *Taeniopygia guttata*) assessed twice were repeatable ($r = 0.84$, $P < 0.001$, $n = 43$). Units are reported as minutes.

## Plasma antioxidants

The total antioxidant status (TAS) of blood plasma was analyzed to estimate the availability of circulating hydrosoluble antioxidants. Since the idea that this measure assesses all the antioxidants is questionable, the term "total" was avoided, and hence, we will only use the generic "Plasma Antioxidants" (PLAOX). The procedure is based on *Miller et al. (1993)* modified by *Cohen, Klasing & Ricklefs (2007)* and *Romero-Haro & Alonso-Alvarez (2014)*. Repeatability calculated on other samples of red-legged partridges assessed twice was high ($r = 0.94$, $P < 0.001$, $n = 20$; *Galván & Alonso-Alvarez, 2009*).

## Plasma biochemistry

Albumin, uric acid, triglycerides, LDL-cholesterol and total cholesterol levels in plasma were determined with commercial kits (Biosystems SA, Barcelona, Spain) with an automated spectrophotometer (A25-Autoanalyzer; Biosystems SA, Barcelona, Spain). The last three parameters are components of lipoproteins that act as carotenoid carriers in blood (*McGraw & Parker, 2006*). They were assessed to test for differences in lipid absorption due to direct diquat effects on the gut (see also *Alonso-Alvarez & Galván, 2011*), but the diquat factor or its interaction with the CAR factor did not provide any significant influence on their levels (all *P*-values > 0.16).

## Lipid peroxidation

The measurement of lipid peroxidation in plasma, liver and heart was carried out following the method described in *Romero-Haro & Alonso-Alvarez (2014)*. Livers and hearts were previously diluted (1:10 w/v) and were homogenized with a stock buffer (phosphate buffer 0.01 M adjusted to pH 7.4 with HCl 37%). Aliquots of 50 μL of the samples (plasma, homogenized liver and heart samples, and standards) were then capped and vortexed for 5 s, and were analyzed as described in *Romero-Haro & Alonso-Alvarez (2014)*. Zebra finch plasma samples assessed twice provided very high within-session ($r = 0.97$, $n = 20$, $P < 0.001$) and between-session ($r = 0.98$, $n = 20$, $P < 0.001$) repeatabilities (*Romero-Haro & Alonso-Alvarez, 2014*).

## Statistical analyses

All the analyses were performed using SAS v9.3 software (SAS Institute, Carry North Carolina, USA). The analyses are organized in two parts: (1) one testing the influence of carotenoid supplements only, and (2) the second analyzing the impact of the oxidative challenge (diquat exposure) and its interaction with carotenoid treatments.

The treatment effects on the number of birds producing eggs were calculated from contingency tables ($\chi^2$). These analyses were separately performed for each experimental period (carotenoid exposure only or diquat exposure) and sex. Sex was considered

because some females escaped during the experiment and hence sample sizes differed between sexes (see above). The variability in the number of eggs per individual was tested using a GENMOD procedure in the SAS software, including the number of eggs as a multinomial variable with cumulative logit link.

To test the carotenoid treatment (CAR hereafter) effect on color and blood variables throughout the study (i.e., three different measures), repeated-measures mixed models (PROC MIXED in SAS; *Littell et al., 2006*) were used. In these models, the sampling event (TIME hereafter) was included as the repeated-measures factor, whereas the identity of the individual nested into cage identity was the subject term (REPEATED statement; *Littell et al., 2006*). CAR (four-level factor), TIME (three-level factor) and sex were always included in the models as fixed effects, testing their two- and three-way interactions. Since the aim was exclusively testing the CAR effect with the highest available statistical power, these repeated-measure models did not include data from those individuals exposed to diquat (day 82 only).

To analyze the effect of diquat, variability at the last sampling (day 82) was analyzed by generalized mixed models (PROC MIXED in SAS). Here, CAR and diquat treatments and sex were tested as fixed factors, testing their interactions. Color and blood levels at the precedent sampling event (day 48) were tested as covariates to correct for subtle differences between groups at the start of the diquat exposure (see section Variability after diquat exposure).

Other different covariates were added to the models. Thus, as previously mentioned, the redness (inverse of hue) of the eye ring and bill was controlled for total brightness. In the case of the leg, the redness of the red chip was tested. In all the repeated-measures mixed models testing the CAR effect, the influence of plasma vitamin (tocopherol and retinol) levels was tested by including them as covariates. In all the mixed models testing the diquat effect, plasma vitamin levels in the last sampling event, as well as vitamin levels in every internal tissue and ornaments, were also added. In models testing plasma MDA values, plasma triglyceride levels were added to control for potential influences of lipid variability in the blood (*Romero-Haro & Alonso-Alvarez, 2014*; *Romero-Haro, Canelo & Alonso-Alvarez, 2015*). In models testing PLAOX, uric acid, and albumin values were simultaneously tested to control for influences of recent food intake (*Cohen, Klasing & Ricklefs, 2007*). To control for subtle differences in reproductive investment, the number of eggs produced at the end of each sampling interval ("eggs") was also tested as a covariate in repeated models (Table 2). In models testing final variability (Tables 3 and 4), the total number of eggs at the end of the study or the number of eggs during only the diquat experiment were tested as alternative covariates (in different models). The lag time (min) to start hemolysis and hematocrit were added as covariates in models testing resistance to hemolysis. Finally, the identity of the bird nested into the identity of the cage and the laboratory session were included as random factors (*P*-values ranging from < 0.001 to 0.476).

All the mixed models were explored from the saturated models. They firstly included all the covariates (although see alternative options above), fixed factors, and factor interactions. Alternative models were then tested by removing terms at *P* > 0.10 by

**Table 2  Mixed models testing the interaction between carotenoid treatment and time.** The reported tests are the best fitted models with an interaction at $P < 0.10$, or instead, when it is removed at higher $P$-values by following a backward-step wise procedure (see Methods).

| Dependent variable | Terms in the model | Slope | SE | F | df | P |
|---|---|---|---|---|---|---|
| Eye rings redness | Carotenoid | | | 10.22 | 3,167 | <0.001 |
| | Sex | | | 3.97 | 1,167 | 0.048 |
| | Time | | | 5.43 | 2,237 | 0.005 |
| | Carotenoid × time | | | 2.65 | 6,237 | 0.017 |
| | Eggs | −0.001 | 0.001 | 5.05 | 1,237 | 0.026 |
| | Total brightness | −0.0001 | 0.0001 | 18.05 | 1,237 | <0.001 |
| | Plasma tocopherol | 0.026 | 0.009 | 7.98 | 1,237 | 0.005 |
| Bill redness | Carotenoid | | | 6.58 | 3,168 | <0.001 |
| | Time | | | 8.64 | 3,235 | <0.001 |
| | Carotenoid × time | | | 1.96 | 6,235 | 0.072 |
| | Eggs | −0.001 | 0.001 | 5.87 | 1,235 | 0.016 |
| | Total brightness | −0.0002 | 0.00002 | 63.68 | 1,235 | <0.001 |
| | Plasma tocopherol | 0.033 | 0.011 | 9.23 | 1,235 | 0.003 |
| Legs redness | Carotenoid | | | 6.04 | 3,167 | <0.001 |
| | Sex | | | 17.6 | 1,167 | <0.001 |
| | Time | | | 12.44 | 2,227 | <0.001 |
| | Sex × time | | | 1.36 | 2,227 | 0.258 |
| | Carotenoid × time | | | 0.63 | 6,227 | 0.703 |
| | Eggs | −0.025 | 0.022 | 1.28 | 1,227 | 0.259 |
| | Red chip | 1.251 | 0.274 | 20.85 | 1,227 | <0.001 |
| | Plasma tocopherol | 1.129 | 0.467 | 5.84 | 1,227 | 0.017 |
| | Plasma retinol | −1.854 | 1.147 | 2.61 | 1,227 | 0.108 |
| Plasma lutein | Carotenoid | | | 105.1 | 3,164 | <0.001 |
| | Sex | | | 54.22 | 1,164 | <0.001 |
| | Time | | | 69.61 | 2,237 | <0.001 |
| | Sex × carotenoid | | | 5.22 | 3,164 | 0.002 |
| | Carotenoid × time | | | 36.81 | 6,237 | <0.001 |
| | Plasma tocopherol | 0.456 | 0.026 | 313.86 | 1,237 | <0.001 |
| | Plasma retinol | 0.168 | 0.065 | 6.74 | 1,237 | 0.01 |
| | Eggs | −0.004 | 0.001 | 8.57 | 1,237 | 0.004 |
| Plasma zeaxanthin | Carotenoid | | | 309.6 | 3,164 | <0.001 |
| | Sex | | | 47.35 | 1,164 | <0.001 |
| | Time | | | 73.54 | 2,235 | <0.001 |
| | Carotenoid × time | | | 95.93 | 6,235 | <0.001 |
| | Sex × carotenoid | | | 3.68 | 3,164 | 0.013 |
| | Sex × time | | | 4.4 | 2,235 | 0.013 |
| | Plasma tocopherol | 0.379 | 0.026 | 216.34 | 1,235 | <0.001 |
| | Plasma retinol | 0.188 | 0.064 | 8.72 | 1,235 | 0.004 |
| | Eggs | −0.006 | 0.001 | 20.9 | 1,235 | <0.001 |

| Dependent variable | Terms in the model | Slope | SE | F | df | P |
|---|---|---|---|---|---|---|
| Plasma tocopherol | Carotenoid | | | 2.61 | 3,167 | 0.053 |
| | Sex | | | 2.56 | 1,167 | 0.112 |
| | Time | | | 117.68 | 2,236 | <0.001 |
| | Carotenoid × time | | | 2.63 | 6,236 | 0.017 |
| | Sex × time | | | 4.89 | 2,236 | 0.008 |
| | Plasma retinol | 0.548 | 0.122 | 20.25 | 1,236 | <0.001 |
| | Eggs | −0.009 | 0.002 | 14.94 | 1,236 | <0.001 |
| Plasma retinol | Carotenoid | | | 2.11 | 3,168 | 0.101 |
| | Time | | | 36.06 | 2,238 | <0.001 |
| | Carotenoid × time | | | 1.01 | 6,238 | 0.421 |
| | Plasma tocopherol | 0.0852 | 0.019 | 20.8 | 1,238 | <0.001 |
| | Eggs | −0.002 | 0.001 | 5.14 | 1,238 | 0.024 |
| UA & ALB-corrected PLAOX | Carotenoid | | | 7.19 | 3,164 | <0.001 |
| | Time | | | 0.38 | 2,187 | 0.686 |
| | Carotenoid × time | | | 2.57 | 6,187 | 0.021 |
| | Uric acid | 0.768 | 0.053 | 2.14 | 1,187 | <0.001 |
| | Albumin | −0.498 | 0.112 | 19.99 | 1,187 | <0.001 |
| | Plasma retinol | 0.228 | 0.098 | 5.17 | 1,187 | 0.024 |
| Plasma TRG-corrected MDA | Carotenoid | | | 1.05 | 3,164 | 0.371 |
| | Sex | | | 0.29 | 1,164 | 0.593 |
| | Time | | | 14.43 | 2,228 | <0.001 |
| | Carotenoid × time | | | 0.71 | 6,228 | 0.645 |
| | Sex × carotenoid | | | 1.8 | 3,164 | 0.149 |
| | Sex × time | | | 0.78 | 2,228 | 0.46 |
| | Plasma tocopherol | −0.04 | 0.045 | 0.81 | 1,228 | 0.37 |
| | Plasma retinol | −0.089 | 0.108 | 0.67 | 1,228 | 0.413 |
| | Plasma triglycerides | 0.298 | 0.031 | 90.13 | 1,228 | <0.001 |
| | Eggs | 0.001 | 0.002 | 0.37 | 1,228 | 0.543 |
| Resistance to oxidative stress in erythrocytes | Carotenoid | | | 1.48 | 3,163 | 0.223 |
| | Sex | | | 1.3 | 1,163 | 0.256 |
| | Time | | | 1.7 | 2,203 | 0.185 |
| | Carotenoid × time | | | 0.69 | 6,203 | 0.657 |
| | Sex × carotenoid | | | 0.59 | 3,163 | 0.619 |
| | Plasma retinol | −20.202 | 8.811 | 5.26 | 1,203 | 0.023 |
| | Eggs | −0.339 | 0.18 | 3.55 | 1,203 | 0.061 |
| | Lag time | −0.122 | 0.021 | 35.9 | 1,203 | <0.001 |

**Note:**
ALB, albumin; MDA, malondyaldehydes; PLAOX, plasma antioxidants; TRG, tryglycerides; UA, uric acid.

**Table 3 Mixed models testing how the exposure to oxidative stress (diquat) interacted with the dietary carotenoid treatment at the end of the experiment.** The level of each dependent variable in the sampling event precedent to the diquat exposure is included as a covariate for color and blood variables. The models describe the backward step (using the $P = 0.10$ threshold) previous to remove the diquat × CAR interaction (i.e. when it was non-significant; Methods).

| Dependent variable | Terms in the model | Slope | SE | F | df | P |
|---|---|---|---|---|---|---|
| Eye rings redness | Carotenoid | | | 24.93 | 3,147 | <0.001 |
| | Diquat | | | 0.03 | 1,146 | 0.859 |
| | Sex | | | 0.8 | 1,146 | 0.373 |
| | Carotenoid × diquat | | | 1.38 | 3,146 | 0.252 |
| | Sex × diquat | | | 3.21 | 1,146 | 0.075 |
| | Total brightness | −0.00004 | 0.00002 | 6.25 | 1,147 | 0.014 |
| | Eye ring redness in day 48 | 0.291 | 0.064 | 20.95 | 1,147 | <0.001 |
| | Liver vitamin A | 0.0004 | 0.0003 | 1.33 | 1,147 | 0.251 |
| | Eye ring tocopherol | 0.025 | 0.012 | 4.43 | 1,146 | 0.037 |
| | Eggs during diquat experiment | −0.002 | 0.0008 | 4.44 | 1,146 | 0.037 |
| Bill redness | Carotenoid | | | 17.19 | 383.2 | <0.001 |
| | Diquat | | | 4.23 | 176.6 | 0.043 |
| | Sex | | | 1.46 | 175.9 | 0.231 |
| | Carotenoid × diquat | | | 1.03 | 374.3 | 0.382 |
| | Sex × carotenoid | | | 1.34 | 369.7 | 0.269 |
| | Total brightness | −0.0001 | 0.00003 | 23.14 | 1,143 | <0.001 |
| | Bill redness in day 48 | 0.151 | 0.056 | 7.4 | 1,134 | 0.007 |
| | Bill tocopherol | 0.048 | 0.014 | 11.7 | 1,118 | <0.001 |
| | Liver vitamin A | 0.001 | 0.0004 | 5.72 | 1,138 | 0.018 |
| Leg redness | Carotenoid | | | 10.55 | 3,136 | <0.001 |
| | Diquat | | | 0.23 | 1,137 | 0.631 |
| | Sex | | | 3.98 | 1,138 | 0.048 |
| | Carotenoid × diquat | | | 0.67 | 3,137 | 0.575 |
| | Sex × diquat | | | 0.33 | 1,137 | 0.567 |
| | Sex × carotenoid | | | 1.69 | 3,137 | 0.173 |
| | Red chip | 1.018 | 0.221 | 21.28 | 128.8 | <0.001 |
| | Leg redness in day 48 | 0.594 | 0.065 | 82.46 | 1,137 | <0.001 |
| | Leg tocopherol | 2.752 | 0.605 | 20.73 | 1,137 | <0.001 |
| | Liver tocopherol | −1.469 | 0.544 | 7.28 | 1,138 | 0.008 |
| | Liver vitamin A | −0.024 | 0.015 | 2.73 | 1,135 | 0.101 |
| | Total number of eggs | −0.032 | 0.014 | 5.1 | 1,137 | 0.026 |
| Total astaxanthin in the eye rings | Carotenoid | | | 51.64 | 3,146 | <0.001 |
| | Diquat | | | 1.47 | 1,151 | 0.227 |
| | Carotenoid × diquat | | | 3.21 | 3,147 | 0.025 |
| | Sex | | | 17.4 | 1,148 | <0.001 |
| | Sex × carotenoid | | | 3.21 | 3,146 | 0.025 |
| | Tocopherol in the eye rings | 0.674 | 0.06 | 125.22 | 1,149 | <0.001 |
| | Total number of eggs | −0.004 | 0.002 | 6.54 | 1,145 | 0.012 |

| Dependent variable | Terms in the model | Slope | SE | F | df | P |
|---|---|---|---|---|---|---|
| Total papilioerythrinone in the eye rings | Carotenoid | | | 19.9 | 388.3 | <0.001 |
| | Diquat | | | 0.55 | 180.1 | 0.46 |
| | Sex | | | 13.83 | 179.6 | <0.001 |
| | Carotenoid × diquat | | | 0.33 | 377.4 | 0.804 |
| | Sex × diquat | | | 1.74 | 181.8 | 0.19 |
| | Sex × carotenoid | | | 3.71 | 379.5 | 0.015 |
| | Fat retinol | 0.032 | 0.031 | 1.04 | 1,136 | 0.309 |
| | Plasma tocopherol | 0.3 | 0.182 | 2.71 | 1,140 | 0.102 |
| | Tocopherol in the eye rings | 0.85 | 0.149 | 32.42 | 1,140 | <0.001 |
| | Total number of eggs | −0.007 | 0.004 | 3.03 | 183.8 | 0.086 |
| Tocopherol in the eye rings | Carotenoid | | | 3.64 | 371.7 | 0.017 |
| | Diquat | | | 1.22 | 175.7 | 0.272 |
| | Carotenoid × diquat | | | 2.63 | 373.5 | 0.056 |
| | Total number of eggs | −0.006 | 0.002 | 8.96 | 171.1 | 0.004 |
| Total astaxanthin in the bill | Carotenoid | | | 141.3 | 3,151 | <0.001 |
| | Sex | | | 5.43 | 1,155 | 0.021 |
| | Diquat | | | 4.68 | 1,157 | 0.032 |
| | Carrotenoid × diquat | | | 2.67 | 3,155 | 0.049 |
| | Plasma tocopherol | 1.176 | 0.061 | 371.2 | 1,158 | <0.001 |
| | Total number of eggs | −0.007 | 0.002 | 17.86 | 1,151 | <0.001 |
| Total papilioerythrinone in the bill | Carotenoid | | | 134.5 | 364.8 | <0.001 |
| | Diquat | | | 0.08 | 168.8 | 0.774 |
| | Sex | | | 2.66 | 175.8 | 0.107 |
| | Carotenoid × diquat | | | 1.76 | 366 | 0.163 |
| | Tocopherol in the bill | 1.536 | 0.133 | 133.35 | 1,134 | <0.001 |
| | Plasma retinol | −9.373 | 4.689 | 4 | 1,140 | 0.048 |
| | Total number of eggs | −0.009 | 0.003 | 9.17 | 173.4 | 0.003 |
| Tocopherol in the bill | Carotenoid | | | 2.94 | 3,158 | 0.035 |
| | Diquat | | | 3.91 | 1,162 | 0.05 |
| | Carotenoid × diquat | | | 3.09 | 3,160 | 0.029 |
| | Sex | | | 5.6 | 1,161 | 0.019 |
| | Total number of eggs | −0.007 | 0.002 | 9.66 | 1,159 | 0.002 |
| Total astaxanthin in the legs | Carotenoid | | | 7.36 | 392.1 | <0.001 |
| | Diquat | | | 0.13 | 178.8 | 0.7168 |
| | Sex | | | 2.98 | 186.9 | 0.088 |
| | Carotenoid × diquat | | | 0.07 | 376 | 0.974 |
| | Sex × diquat | | | 0.14 | 176.6 | 0.712 |
| | Sex × carotenoid | | | 0.56 | 374.3 | 0.645 |
| | Plasma tocopherol | 0.146 | 0.113 | 1.66 | 1,144 | 0.199 |
| | Liver vitamin A | 0.004 | 0.002 | 3.32 | 1,133 | 0.071 |
| | Fat retinol | −0.032 | 0.018 | 3.19 | 1,130 | 0.077 |
| | Tocopherol in the legs | 0.526 | 0.097 | 29.25 | 1,141 | <0.001 |
| | Total number of eggs | 0.003 | 0.002 | 1.37 | 187.5 | 0.244 |

(Continued)

| Dependent variable | Terms in the model | Slope | SE | F | df | P |
|---|---|---|---|---|---|---|
| Total papilioerythrinone in the legs | Carotenoid | | | 4.17 | 392.9 | 0.008 |
| | Diquat | | | 0.61 | 184.9 | 0.436 |
| | Sex | | | 6.93 | 191.3 | 0.01 |
| | Carotenoid × diquat | | | 0.17 | 382.1 | 0.919 |
| | Sex × diquat | | | 0.24 | 181.1 | 0.627 |
| | Sex × carotenoid | | | 0.16 | 378.8 | 0.919 |
| | Tocopherol in the legs | 0.867 | 0.147 | 34.64 | 1,143 | <0.001 |
| | Liver vitamin A | −0.003 | 0.004 | 0.49 | 1,144 | 0.484 |
| Tocopherol in the legs | Carotenoid | | | 4.09 | 382.2 | 0.009 |
| | Diquat | | | 2.4 | 183.8 | 0.125 |
| | Carotenoid × diquat | | | 1.21 | 182.4 | 0.3126 |
| | Sex | | | 5.98 | 189.5 | 0.016 |
| | Total number of eggs | −0.004 | 0.002 | 4.41 | 182.1 | 0.039 |
| Plasma lutein | Carotenoid | | | 151.01 | 3,149 | <0.001 |
| | Diquat | | | 0.01 | 1,149 | 0.925 |
| | Carotenoid × diquat | | | 2.84 | 3,149 | 0.04 |
| | Lutein at day 48 | 0.446 | 0.062 | 51.81 | 1,149 | <0.001 |
| | Plasma tocopherol | 0.479 | 0.0326 | 215.52 | 1,149 | <0.001 |
| | Eggs during diquat experiment | −0.003 | 0.002 | 4.42 | 1,149 | 0.037 |
| Plasma zeaxanthin | Carotenoid | | | 321.37 | 3,146 | <0.001 |
| | Diquat | | | 0.83 | 1,146 | 0.363 |
| | Carotenoid × diquat | | | 1.57 | 3,146 | 0.2 |
| | Zeaxanthin at day 48 | 0.307 | 0.07 | 19.24 | 1,146 | <0.001 |
| | Plasma tocopherol | 0.572 | 0.039 | 219.4 | 1,146 | <0.001 |
| | Fat tocopherol | −0.037 | 0.013 | 8.24 | 1,146 | 0.005 |
| | Liver vitamin A | 0.002 | 0.001 | 4.79 | 1,146 | 0.03 |
| | Eggs during diquat experiment | −0.004 | 0.002 | 4.05 | 1,146 | 0.046 |
| Plasma tocopherol | Carotenoid | | | 5.26 | 395.1 | 0.002 |
| | Diquat | | | 2.72 | 182.3 | 0.103 |
| | Sex | | | 0.14 | 186.2 | 0.71 |
| | Carotenoid × diquat | | | 0.7 | 379.3 | 0.552 |
| | Sex × diquat | | | 0.7 | 182.7 | 0.404 |
| | Tocopherol at day 48 | 0.199 | 0.069 | 8.31 | 1,139 | 0.005 |
| | Liver vitamin A | −0.004 | 0.002 | 3.99 | 1,134 | 0.048 |
| | Fat retinol | 0.03 | 0.016 | 3.34 | 1,138 | 0.07 |
| | Plasma retinol | 0.259 | 0.162 | 2.57 | 1,141 | 0.111 |
| | Total number of eggs | −0.003 | 0.002 | 2.21 | 184.2 | 0.141 |
| Plasma retinol | Carotenoid | | | 1.36 | 380.5 | 0.262 |
| | Diquat | | | 1.54 | 180.3 | 0.218 |
| | Sex | | | 0.51 | 177.9 | 0.476 |
| | Carotenoid × diquat | | | 0.51 | 377.7 | 0.678 |

| Dependent variable | Terms in the model | Slope | SE | F | df | P |
|---|---|---|---|---|---|---|
| | Sex × diquat | | | 0.31 | 178.7 | 0.579 |
| | Sex × carotenoid | | | 0.64 | 379.5 | 0.589 |
| | Retinol at day 48 | 33.86 | 4.27 | 62.89 | 1,134 | <0.001 |
| | Plasma tocopherol | 5.209 | 1.95 | 7.13 | 1,140 | 0.009 |
| | Eggs during diquat experiment | −0.258 | 0.095 | 7.34 | 173.6 | 0.008 |
| Liver lutein | Carotenoid | | | 128.63 | 3,157 | <0.001 |
| | Diquat | | | 0.06 | 1,157 | 0.811 |
| | Carotenoid × diquat | | | 1.49 | 3,157 | 0.22 |
| | Plasma tocopherol | 0.075 | 0.036 | 4.45 | 1,157 | 0.037 |
| | Liver tocopherol | 0.263 | 0.028 | 85.5 | 1,157 | <0.001 |
| Liver zeaxanthin | Carotenoid | | | 315.42 | 3,151 | <0.001 |
| | Diquat | | | 0 | 1,154 | 0.971 |
| | Carotenoid × diquat | | | 3.06 | 3,151 | 0.03 |
| | Plasma tocopherol | 0.1 | 0.046 | 4.69 | 1,151 | 0.031 |
| | Liver tocopherol | 0.341 | 0.04 | 74.08 | 140.3 | <0.001 |
| | Plasma retinol | 0.003 | 0.001 | 3.6 | 1,153 | 0.06 |
| Liver tocopherol | Carotenoid | | | 12.77 | 3,161 | <0.001 |
| | Diquat | | | 6.47 | 1,161 | 0.012 |
| | Carotenoid × diquat | | | 2.76 | 3,161 | 0.044 |
| Liver vitamin A | Carotenoid | | | 57.35 | 3,152 | <0.001 |
| | Diquat | | | 0.04 | 1,154 | 0.834 |
| | Sex | | | 22.3 | 1,154 | <0.001 |
| | Carotenoid × diquat | | | 0.47 | 3,152 | 0.707 |
| | Sex × diquat | | | 0.71 | 1,152 | 0.399 |
| | Plasma tocopherol | −11.032 | 4.001 | 7.6 | 1,153 | 0.007 |
| | Liver tocopherol | 10.309 | 3.538 | 8.49 | 155.8 | 0.005 |
| | Total number of eggs | −0.394 | 0.07 | 31.26 | 1,152 | <0.001 |
| Fat lutein | Carotenoid | | | 12.87 | 3,147 | <0.001 |
| | Diquat | | | 0.06 | 1,148 | 0.808 |
| | Sex | | | 0.13 | 1,147 | 0.716 |
| | Carotenoid × diquat | | | 0.21 | 3,148 | 0.89 |
| | Sex × diquat | | | 0.19 | 1,147 | 0.662 |
| | Sex × carotenoid | | | 1.26 | 3,147 | 0.292 |
| | Fat tocopherol | 1.288 | 0.181 | 50.69 | 1,142 | <0.001 |
| | Liver vitamin A | 0.022 | 0.01 | 4.62 | 1,147 | 0.033 |
| | Plasma retinol | −0.039 | 0.016 | 5.77 | 1,147 | 0.018 |
| | Fat retinol | 0.241 | 0.087 | 7.72 | 1,148 | 0.006 |
| Fat zeaxanthin | Carotenoid | | | 46.74 | 3,148 | <0.001 |
| | Diquat | | | 0.16 | 1,148 | 0.687 |
| | Sex | | | 0.28 | 1,147 | 0.598 |
| | Carotenoid × diquat | | | 0.12 | 3,148 | 0.948 |

(Continued)

| Dependent variable | Terms in the model | Slope | SE | F | df | P |
|---|---|---|---|---|---|---|
| | Sex × diquat | | | 0.88 | 1,148 | 0.349 |
| | Sex × carotenoid | | | 1.23 | 3,148 | 0.302 |
| | Fat tocopherol | 0.911 | 0.15 | 37.07 | 1,129 | <0.001 |
| | Liver vitamin A | 0.019 | 0.01 | 4.55 | 1,148 | 0.035 |
| | Plasma retinol | −0.023 | 0.014 | 3.01 | 1,148 | 0.085 |
| | Fat retinol | 0.203 | 0.073 | 7.84 | 1,148 | 0.006 |
| Fat tocopherol | Carotenoid | | | 0.57 | 395.7 | 0.638 |
| | Diquat | | | 1.3 | 182.8 | 0.257 |
| | Sex | | | 0.04 | 181.4 | 0.849 |
| | Carotenoid × diquat | | | 0.15 | 382 | 0.931 |
| | Sex × diquat | | | 0.31 | 181 | 0.578 |
| | Sex × carotenoid | | | 1.71 | 379.4 | 0.171 |
| | Liver vitamin A | 0.005 | 0.004 | 1.11 | 1,130 | 0.295 |
| | Plasma retinol | −0.012 | 0.007 | 3.05 | 1,138 | 0.083 |
| | Total number of eggs | −0.005 | 0.005 | 0.99 | 191.4 | 0.323 |
| Fat retinol | Carotenoid | | | 29.11 | 3,149 | <0.001 |
| | Diquat | | | 0.18 | 1,148 | 0.671 |
| | Sex | | | 0.14 | 1,147 | 0.706 |
| | Carotenoid × diquat | | | 0.24 | 3,149 | 0.867 |
| | Sex × diquat | | | 0.05 | 1,149 | 0.816 |
| | Sex × carotenoid | | | 1.06 | 3,148 | 0.368 |
| | Plasma tocopherol | 0.64 | 0.535 | 1.43 | 1,149 | 0.234 |
| | Liver tocopherol | 0.136 | 0.428 | 0.1 | 1,149 | 0.752 |
| | Fat tocopherol | −0.103 | 0.165 | 0.39 | 164.6 | 0.537 |
| | Total number of eggs | −0.025 | 0.01 | 6.77 | 1,148 | 0.01 |
| UA & ALB-corrected PLAOX | Carotenoid | | | 3.4 | 379.9 | 0.022 |
| | Diquat | | | 0.37 | 171.4 | 0.543 |
| | Sex | | | 1.61 | 369.7 | 0.209 |
| | Carotenoid × diquat | | | 1.34 | 169.6 | 0.269 |
| | Carotenoid × sex | | | 0.33 | 362.6 | 0.805 |
| | Diquat × sex | | | 0.03 | 175.9 | 0.855 |
| | Carotenoid × diquat × sex | | | 2.85 | 361.3 | 0.045 |
| | AOX at day 48 | 0.331 | 0.11 | 9.07 | 173.1 | 0.004 |
| | Fat tocopherol | 0.054 | 0.03 | 3.21 | 1,103 | 0.076 |
| | Uric acid | 0.056 | 0.005 | 119.03 | 1,101 | <0.001 |
| | Albumin | −0.009 | 0.005 | 3.73 | 1,104 | 0.056 |
| | Liver vitamin A | −0.004 | 0.002 | 3.69 | 187.2 | 0.058 |
| | Eggs during diquat experiment | −0.011 | 0.005 | 4.66 | 183.9 | 0.034 |
| Plasma TRG-corrected MDA | Carotenoid | | | 0.29 | 3,139 | 0.836 |
| | Diquat | | | 6.84 | 1,139 | 0.009 |
| | Sex | | | 4.8 | 1,140 | 0.03 |
| | Carotenoid × diquat | | | 0.86 | 3,139 | 0.466 |

| Dependent variable | Terms in the model | Slope | SE | F | df | P |
|---|---|---|---|---|---|---|
| | Diquat × sex | | | 4.45 | 1,140 | 0.037 |
| | TRG-corrected MDA at day 48 | 2.419 | 0.745 | 10.48 | 1,139 | 0.002 |
| | Triglycerides | 0.004 | 0.001 | 36.4 | 1,140 | <0.001 |
| | Eggs during diquat experiment | 0.126 | 0.044 | 8.28 | 1,140 | 0.005 |
| Liver MDA | Carotenoid | | | 1.27 | 388 | 0.289 |
| | Diquat | | | 0.2 | 177.4 | 0.659 |
| | Sex | | | 22.76 | 180.4 | <0.001 |
| | Carotenoid × diquat | | | 1.96 | 376.7 | 0.127 |
| | Carotenoid × sex | | | 1.21 | 375 | 0.311 |
| | Diquat × sex | | | 0.15 | 175.4 | 0.699 |
| | Carotenoid × diquat × sex | | | 4.65 | 376 | 0.005 |
| | Liver vitamin A | 0.0002 | 0.0001 | 7.07 | 1,138 | 0.009 |
| | Plasma tocopherol | 0.0003 | 0.0001 | 3.73 | 1,139 | 0.055 |
| Heart MDA | Carotenoid | | | 0.09 | 3,157 | 0.963 |
| | Diquat | | | 0.95 | 1,157 | 0.331 |
| | Sex | | | 2.75 | 1,157 | 0.099 |
| | Carotenoid × diquat | | | 1.79 | 3,157 | 0.151 |
| Erythrocyte resistance to oxidative stress | Carotenoid | | | 0.35 | 361.1 | 0.793 |
| | Diquat | | | 5.66 | 161.1 | 0.021 |
| | Carotenoid × diquat | | | 2.27 | 361.4 | 0.09 |
| | Lag time | −0.229 | 0.035 | 44.06 | 1,121 | <0.001 |

**Note:**
ALB, albumin; MDA, malondyaldehydes; PLAOX, plasma antioxidants; TRG, tryglycerides; UA, uric acid.

following a backward-stepwise procedure. The last best-fitted model was also compared to alternatives using the Akaike Information Criteria (AIC), providing similar conclusions. When tested as dependent variables, carotenoids and vitamins were transformed with mathematical functions to attain a normal distribution. All carotenoids and tocopherol levels were log-transformed, whereas vitamin A levels in the liver were transformed by a square root. In subcutaneous fat, carotenoid and retinol levels were standardized into two blocks because some sample sessions gave particularly low values. Differences are always provided as least square means ± SE from models; that is, considering random factors and any term in the final model. Pair-wise comparisons were done by means of LSD post hocs. The description of interactions and their figures in the main text are restricted to tests reporting $P < 0.10$. Other models, figures and tables containing means and SD from raw data are described in Supplemental Information.

## RESULTS

### Egg laying

The treatments did not affect the number of individuals producing eggs during the first (carotenoid supply only; all $\chi^2$ tests: $P > 0.34$) or second (diquat × carotenoid
**Table 4 Best fitted models obtained when the diquat × CAR interaction is removed at *P* > 0.10 after a backward stepwise procedure (see Methods).** Heart MDA and fat tocopherol did not maintain any term (all *P* > 0.10).

| Dependent variable | Terms in the model | Slope | SE | F | df | P |
|---|---|---|---|---|---|---|
| Eye rings redness | Carotenoid | | | 25.55 | 3,154 | <0.001 |
| | Total brightness | −0.00004 | 0.00002 | 7.71 | 1,155 | 0.006 |
| | Eye ring redness in day 48 | 0.286 | 0.064 | 19.99 | 1,155 | <0.001 |
| | Eye ring tocopherol | 0.023 | 0.012 | 6.72 | 1,154 | 0.011 |
| | Eggs during diquat experiment | −0.002 | 0.001 | 6.21 | 1,155 | 0.014 |
| Bill redness | Carotenoid | | | 16.22 | 385.7 | <0.001 |
| | Diquat | | | 4.46 | 177.9 | 0.038 |
| | Total brightness | −0.0001 | 0.00003 | 22.32 | 1,150 | <0.001 |
| | Bill redness in day 48 | 0.163 | 0.054 | 9.1 | 1,141 | 0.003 |
| | Bill tocopherol | 0.045 | 0.013 | 11.6 | 1,132 | <0.001 |
| | Liver vitamin A | 0.0008 | 0.0004 | 4.26 | 1,141 | 0.041 |
| Leg redness | Carotenoid | | | 10.86 | 377.2 | <0.001 |
| | Sex | | | 2.86 | 180.3 | 0.095 |
| | Red chip | 1.035 | 0.217 | 22.83 | 127.4 | <0.001 |
| | Leg redness in day 48 | 0.58 | 0.064 | 81.19 | 1,147 | <0.001 |
| | Leg tocopherol | 3.022 | 0.589 | 26.36 | 1,146 | <0.001 |
| | Liver tocopherol | −1.78 | 0.523 | 11.59 | 1,147 | <0.001 |
| Total papilioerythrinone in the eye rings | Carotenoid | | | 25.34 | 3,153 | <0.001 |
| | Sex | | | 15.53 | 1,156 | <0.001 |
| | Tocopherol in the eye ring | 0.953 | 0.14 | 46.43 | 1,158 | <0.001 |
| | Eggs (total) | −0.009 | 0.004 | 5.43 | 1,153 | 0.021 |
| Total papilioerythrinone in the bill | Carotenoid | | | 131.19 | 368.3 | <0.001 |
| | Sex | | | 2.9 | 176.3 | 0.092 |
| | Tocopherol in the bill | 1.564 | 0.129 | 147.34 | 1,142 | <0.001 |
| | Plasma retinol | −9.038 | 4.666 | 3.75 | 1,145 | 0.055 |
| | Eggs (total) | −0.009 | 0.003 | 8.37 | 177.9 | 0.005 |
| Total astaxanthin in the legs | Carotenoid | | | 9.4 | 377.9 | <0.001 |
| | Sex | | | 10.56 | 187 | 0.002 |
| | Tocopherol in the legs | 0.564 | 0.081 | 48.28 | 1,159 | <0.001 |
| Total papilioerythrinone in the legs | Carotenoid | | | 5.24 | 384.2 | 0.002 |
| | Sex | | | 7.39 | 186.3 | 0.008 |
| | Tocopherol in the legs | 0.866 | 0.139 | 38.79 | 1,152 | <0.001 |
| Tocopherol in the legs | Carotenoid | | | 3.97 | 385.7 | 0.011 |
| | Sex | | | 5.74 | 189.2 | 0.019 |
| | Eggs (total) | −0.004 | 0.002 | 4.93 | 185.4 | 0.029 |
| Plasma zeaxanthin | Carotenoid | | | 322.78 | 3,150 | <0.001 |
| | Zeaxanthin in day 48 | 0.31 | 0.07 | 19.6 | 3,150 | <0.001 |
| | Plasma tocopherol | 0.561 | 0.038 | 217.91 | 3,150 | <0.001 |
| | Fat tocopherol | −0.038 | 0.013 | 8.72 | 3,150 | 0.004 |
| | Liver vitamin A | 0.002 | 0.001 | 5.35 | 3,150 | 0.022 |
| | Eggs during diquat experiment | −0.004 | 0.002 | 4.06 | 3,150 | 0.046 |

| Dependent variable | Terms in the model | Slope | SE | F | df | P |
|---|---|---|---|---|---|---|
| Plasma tocopherol | Carotenoid | | | 5.78 | 398.2 | 0.001 |
| | Diquat | | | 4.26 | 186.9 | 0.042 |
| | Tocopherol in day 48 | 0.19 | 0.066 | 8.32 | 1,144 | 0.005 |
| | Liver vitamin A | −0.003 | 0.001 | 2.89 | 1,147 | 0.091 |
| | Fat retinol | 0.034 | 0.016 | 4.81 | 1,147 | 0.029 |
| Plasma retinol | Plasma tocopherol | 5.608 | 2.058 | 7.43 | 1,153 | 0.007 |
| | Eggs during diquat experiment | −0.278 | 0.112 | 6.18 | 178 | 0.015 |
| Liver lutein | Carotenoid | | | 130.26 | 3,161 | <0.001 |
| | Plasma tocopherol | 0.071 | 0.035 | 3.98 | 1,161 | 0.048 |
| | Liver tocopherol | 0.263 | 0.028 | 87 | 1,161 | <0.001 |
| Liver vitamin A | Carotenoid | | | 59.87 | 3,157 | <0.001 |
| | Sex | | | 23.03 | 1,159 | <0.001 |
| | Plasma tocopherol | −11.396 | 3.947 | 8.34 | 1,157 | 0.004 |
| | Liver tocopherol | 9.738 | 3.418 | 8.12 | 158 | 0.006 |
| | Total number of eggs | −0.396 | 0.069 | 32.78 | 1,157 | <0.001 |
| Fat lutein | Carotenoid | | | 13.3 | 3,156 | <0.001 |
| | Fat tocopherol | 1.29 | 0.175 | 54.41 | 1,153 | <0.001 |
| | Liver vitamin A | 0.02 | 0.01 | 4.42 | 1,157 | 0.037 |
| | Plasma retinol | −0.036 | 0.016 | 5.34 | 1,156 | 0.022 |
| | Fat retinol | 0.271 | 0.084 | 10.44 | 1,157 | 0.002 |
| Fat zeaxanthin | Carotenoid | | | 46.81 | 3,163 | <0.001 |
| | Fat tocopherol | 0.952 | 0.143 | 44.45 | 1,146 | <0.001 |
| | Liver vitamin A | 0.018 | 0.008 | 5.32 | 1,162 | 0.022 |
| | Fat retinol | 0.208 | 0.069 | 9.03 | 1,163 | 0.003 |
| Fat retinol | Carotenoid | | | 35.37 | 3,167 | <0.001 |
| | Total number of eggs | −0.026 | 0.01 | 8.62 | 1,165 | 0.004 |
| Plasma TRG-corrected MDA | Diquat | | | 7.04 | 1,145 | 0.009 |
| | sex | | | 5.13 | 1,147 | 0.025 |
| | Diquat × sex | | | 4.66 | 1,145 | 0.033 |
| | TRG-corrected MDA in day 48 | 2.366 | 0.739 | 10.26 | 1,144 | 0.002 |
| | Triglycerides | 0.004 | 0.001 | 38.25 | 1,146 | <0.001 |
| | Eggs during diquat experiment | 0.136 | 0.043 | 10.17 | 1,146 | 0.002 |
| Resistance to oxidative stress in erythrocytes | Diquat | | | 5.64 | 167.4 | 0.02 |
| | Lag time | −0.242 | 0.034 | 49.91 | 1,128 | <0.001 |

supply interaction; all $\chi^2$ tests: $P > 0.86$) part of the experiment. Similarly, the treatments did not influence the number of eggs produced during the first period (all $\chi^2$ tests: $P > 0.65$) or the total number of eggs laid during the whole study (all $\chi^2$ tests: $P > 0.11$). The addition of tocopherol or retinol values as covariates did not change any of these results. The tests on egg production reported similar results when including those males that were housed with new partners during the study (all tests: $P > 0.10$).

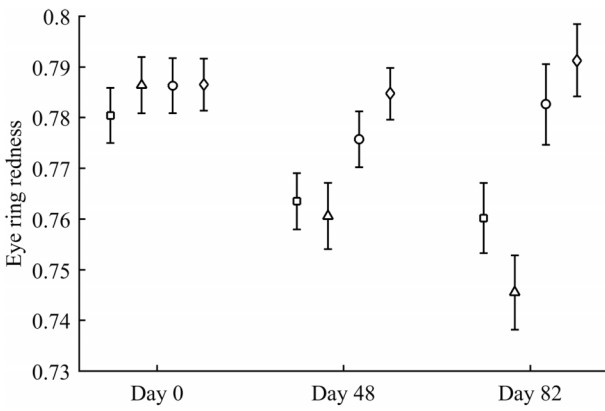

**Figure 1 Changes in eye ring coloration during the experiment depending on the carotenoid treatment.** Least square means ± SE were obtained from the models (see Methods). Squares: Control; Triangles: Astaxanthin; Circles: LutZea; Diamonds: ZeaLut.

## Influence of carotenoid supplements throughout the three sampling events

Body mass was not affected by CAR treatments (time and sex interactions all $P > 0.90$). In contrast, integument coloration changed throughout the study according to carotenoid supplements. Redness decreased throughout reproduction, but the LutZea and ZeaLut groups counteracted this effect (CAR × time interaction) in the eye ring and bill, although the latter trait only showed a trend toward significance (Table 2; Fig. 1). In the eye ring, ZeaLut birds were redder than control and Ast partridges at the second sampling (both $P < 0.05$; Fig. 1). On the last day, LutZea and ZeaLut groups showed redder eye rings than the other treatments ($P < 0.034$), but did not differ between them ($P = 0.411$; Fig. 1). In the bill, differences arose at the last sampling, with LutZea, ZeaLut (both $P < 0.001$) and control (but $P = 0.068$) birds redder than Ast animals. ZeaLut and LutZea birds were also redder than controls, with the latter only a trend ($P = 0.017$ and $0.064$, respectively; LutZea vs. ZeaLut: $P = 0.673$; Table 2; Fig. S1). The legs did not show a significant interaction (Table 2), although ZeaLut birds were redder than controls at the second and last samplings (both $P < 0.013$; Fig. S1).

In terms of plasma pigments, the carotenoid treatment interacted with time (Table 2; Fig. 2). Lutein levels did not differ between ZeaLut and control birds at day 48 ($P = 0.48$), but the other comparisons among groups on that day and at the last sampling were highly significant (all $P < 0.001$), with LutZea birds showing the highest values (Fig. 2). In the case of zeaxanthin, only Ast and control birds did not differ at the last sampling ($P = 0.730$), with the other groups differing clearly (all $P$-values $< 0.001$). Agreeing with predictions, ZeaLut partridges showed the highest zeaxanthin values (Fig. 2).

Plasma vitamins used as covariates in these models (Table 2) were also tested as dependent variables. The CAR × time interaction was not significant for retinol but was for tocopherol (Table 2; Fig. 2). ZeaLut birds showed higher tocopherol values than

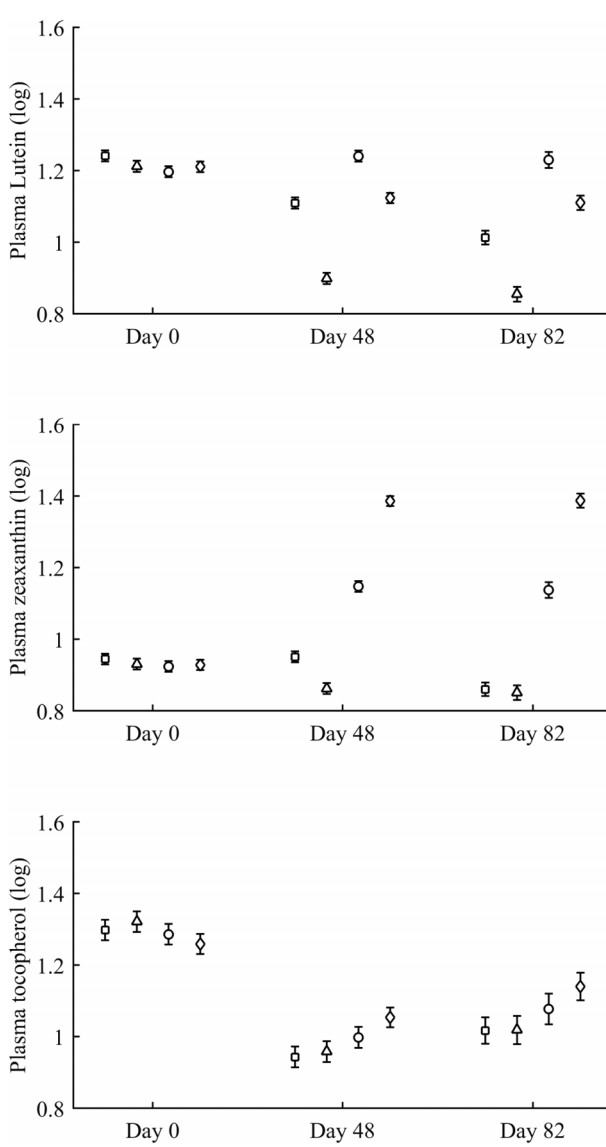

**Figure 2 Changes in plasma carotenoid and tocopherol (log-transformed) levels during the experiment depending on the carotenoid treatment.** Least square means ± SE from the models (see Methods and Table 2). Squares: Control; Triangles: Astaxanthin; Circles: LutZea; Diamonds: ZeaLut.

control and Ast individuals from 48 days to the end of the study (both $P < 0.020$; other comparisons: $P > 0.13$).

PLAOX changed according to the supplemented carotenoid (Table 2; Fig. 3). On day 48, Ast showed higher values than other groups (all $P < 0.012$), with controls reporting higher mean levels than ZeaLut ($P = 0.034$) and LutZea (but $P = 0.098$) birds. At the last sampling, LutZea birds increased their values approaching Ast individuals ($P = 0.715$). Ast birds again differed from the other two groups (both $P < 0.023$), with LutZea animals showing a trend toward higher values than control ($P = 0.052$) and ZeaLut ($P = 0.080$) birds. The interaction remained ($P = 0.020$) when removing albumin and uric acid covariates (Fig. 3).

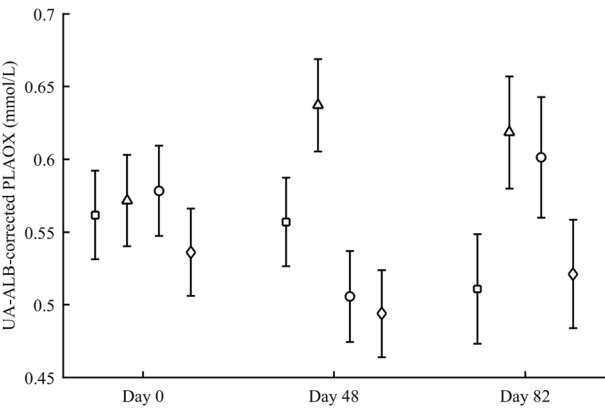

**Figure 3 Changes in the levels (mmol/L) of plasma antioxidant status (controlled for albumin and uric acid levels) during the experiment depending on the carotenoid treatment.** Least square means ± SE from the models (see Methods and Table 2). Squares: Control; Triangles: Astaxanthin; Circles: LutZea; Diamonds: ZeaLut.

Finally, plasma MDA (i.e., corrected or uncorrected for plasma lipid levels) and the resistance to oxidative stress in erythrocytes did not show significant differences with CAR during the study (all $P$-values > 0.64; Table 2).

### Variability after diquat exposure

When testing variables at the end of the study (i.e. after diquat exposure), body mass controlled for tarsus length variability was not influenced by CAR or diquat treatments or their interactions (all $P$ > 0.10). The same was found for circulating LDL and total cholesterol levels (all $P$ > 0.12).

#### *Ornament color and pigments*

In terms of redness, CAR did not clearly interact with diquat in any trait (all $P$ > 0.24; Table 3; Fig. S2). Nonetheless, diquat-exposed birds showed marginally significant redder bills among control and ZeaLut birds ($P$ = 0.051 and 0.084, respectively; Fig. S2). Moreover, in the eye ring model, sex showed a trend toward a significant interaction with diquat ($P$ = 0.069 in its last backward step). Males showed redder eye rings than females, but only among diquat-treated pairs (post hoc: $P$ = 0.020; diquat male: 0.770 ± 0.006; diquat female: 0.757 ± 0.006; control male: 0.762 ± 0.006; control female: 0.764 ± 0.006; other pairwise comparisons: $P$ > 0.18). In any event, in the best-fitted model excluding any interaction (i.e. Table 4), the diquat treatment alone reported a significant effect on bill redness, with diquat-treated birds showing redder bills (Fig. 4).

Best-fitted models for any ornament also showed a strong CAR effect (all $P$-values < 0.001; Table 4). Ast birds were always the palest individuals (all $P$ < 0.001), whereas ZeaLut partridges were the reddest ones, followed by LutZea birds and controls (Fig. 5). Importantly, the difference in color between ZeaLut and LutZea animals was significant in eye rings and legs (both $P$ < 0.044; in the bill: $P$ = 0.065; Fig. 5).

Concerning pigments, neither lutein nor zeaxanthin was detected. Diquat affected astaxanthin levels in the eye ring and bill but depending on CAR (Table 3; Fig. 6). The

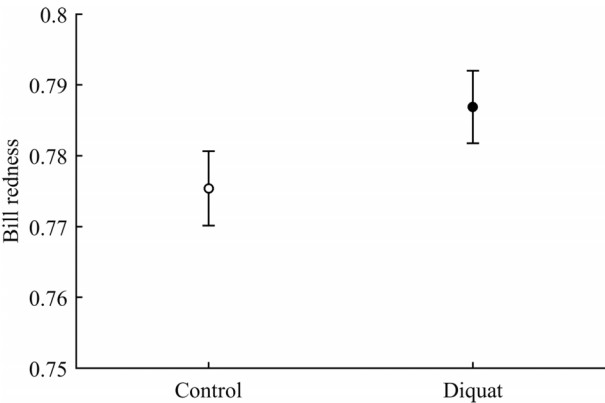

**Figure 4 Effect of the diquat treatment on bill redness.** Least square means ± SE from the models (see Methods and Table 4). Open circles: Control birds; Solid circles: Diquat-treated birds.

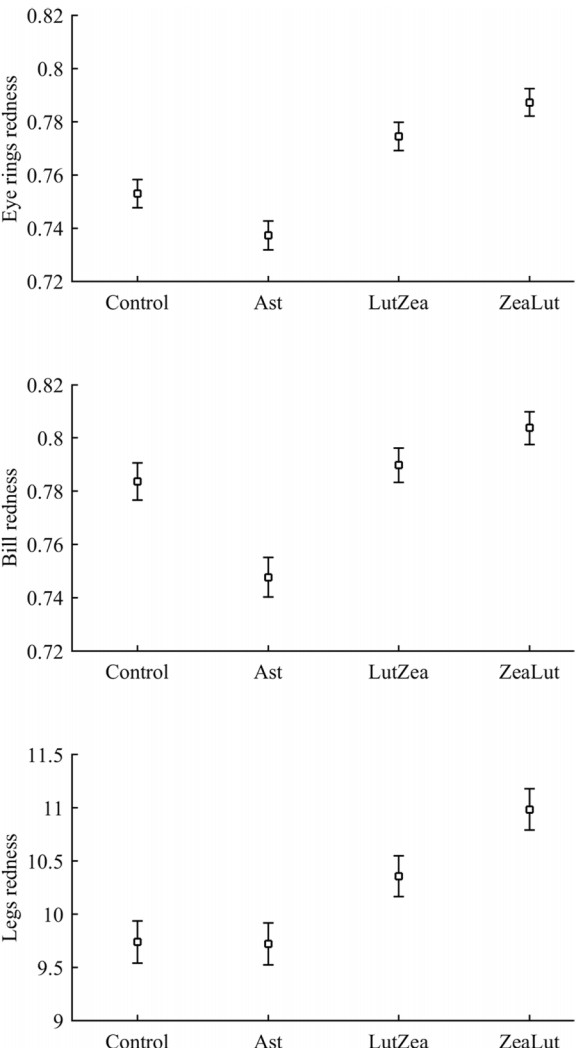

**Figure 5 Final values of ornament coloration depending on carotenoid supplements exclusively.** Least squared means ± SE from the models controlling for the effect of the diquat treatment.

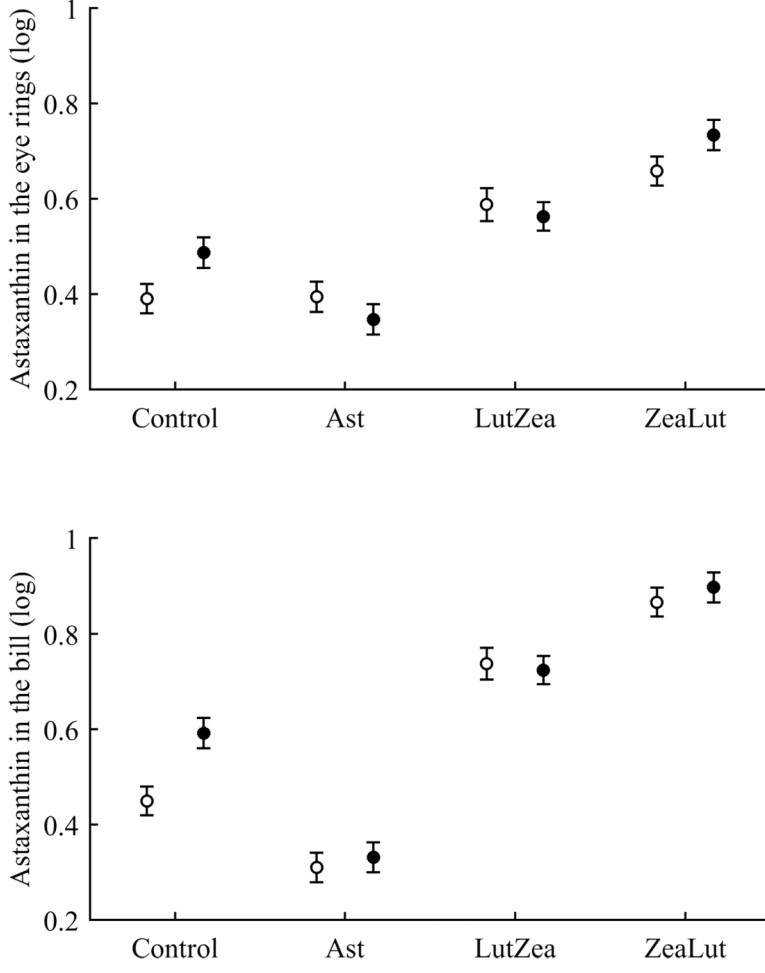

**Figure 6 Levels of astaxanthin in the eye rings and bill after diquat exposure depending on the carotenoid treatment.** Least square means ± SE from the models (see Methods and Table 3). Open circles: Control birds; Solid circles: Diquat-treated birds.

effect was partially due to differences in CAR-controls of both traits (both $P < 0.020$), with diquat-treated birds showing higher astaxanthin concentrations. Nonetheless, in the eye rings ZeaLut birds showed a marginally significant difference in the same direction ($P = 0.057$). In the same eye ring and bill models, all pair-wise comparisons between carotenoid groups (CAR factor: both $P < 0.001$) were significant (all $P < 0.013$), showing increasing astaxanthin values in the following order: Ast, control, LutZea and ZeaLut (Fig. 6). In legs, the diquat × CAR interaction did not affect astaxanthin (Table 3). Only CAR remained in the model (Table 4), with LutZea and ZeaLut birds showing higher astaxanthin levels (Fig. S3) than other groups (all $P < 0.025$), but not differing between them ($P = 0.162$; also Ast vs. control: $P = 0.248$).

In contrast to astaxanthin, papilioerythrinone was unaffected by diquat (any trait: $P > 0.16$; Table 3). The best-fitted model (Table 4) always reported a significant CAR influence (all traits: $P < 0.010$; Fig. S4). In the eye rings, LutZea and ZeaLut birds did not differ ($P = 0.526$), but other comparisons were significant ($P < 0.012$). In the bill, all CAR groups differed ($P < 0.009$), with LutZea showing higher levels than ZeaLut, and

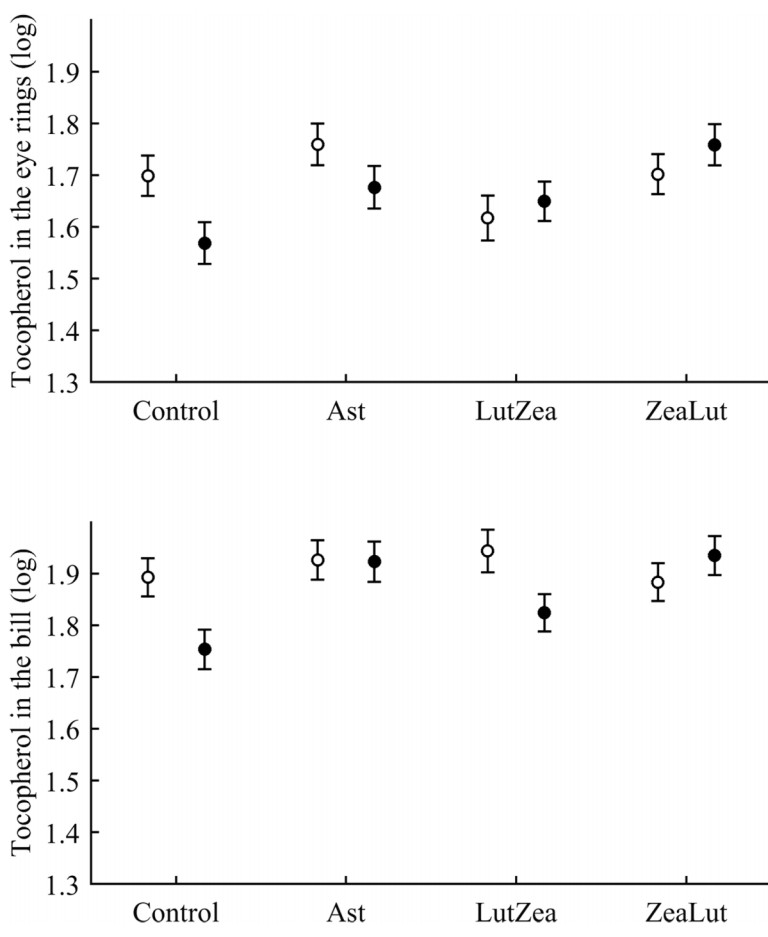

**Figure 7 Levels of tocopherol in the eye rings and bill after diquat exposure depending on the carotenoid treatment.** Least square means ± SE were obtained from the models (see Methods and Table 3). Open circles: Control birds; Solid circles: Diquat-treated birds.

Ast showing the lowest values. In the legs, LutZea presented higher papilioerythrinone levels than other groups (all *P*'s < 0.017; differences among other groups *P* > 0.13; Fig. S4).

Tocopherol and retinol are antioxidant vitamins whose variability could indirectly influence carotenoid values (although tested as covariates in the models; see Methods). Tocopherol, but not retinol, was detected in the ornaments. In the eye ring, the diquat × CAR interaction showed a trend toward significance (*P* = 0.056), with diquat decreasing tocopherol values in controls only (*P* = 0.021; Tables 3 and S2 for raw data; see also Fig. 7). In the same model, the CAR factor (*P* = 0.017) showed that ZeaLut partridges had higher tocopherol levels than LutZea and control birds (both *P* < 0.016), but Ast birds also showed higher vitamin levels than LutZea and control animals (both *P* < 0.039; other comparisons *P* > 0.75).

In the bill, tocopherol was also affected by diquat × CAR (Table 3; Fig. 7). Diquat decreased tocopherol values in control and LutZea individuals (both *P* < 0.05; Fig. 7). The CAR factor (*P* = 0.035) only indicated that controls had lower values than Ast and ZeaLut (both *P* < 0.020). Finally, only the CAR effect was significant in the legs (Tables 3

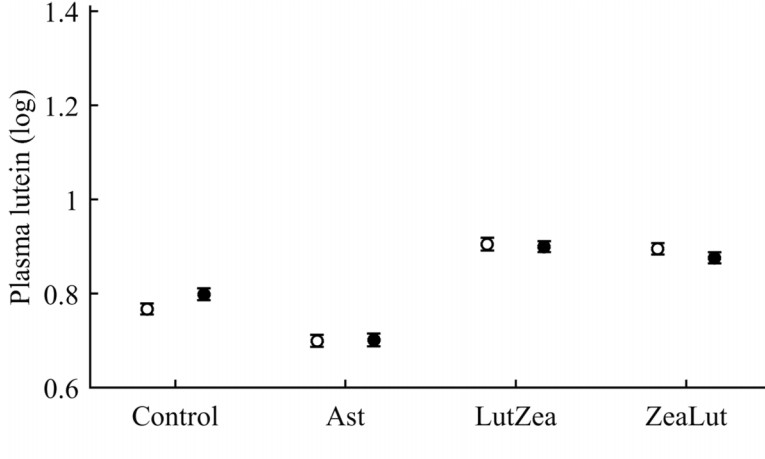

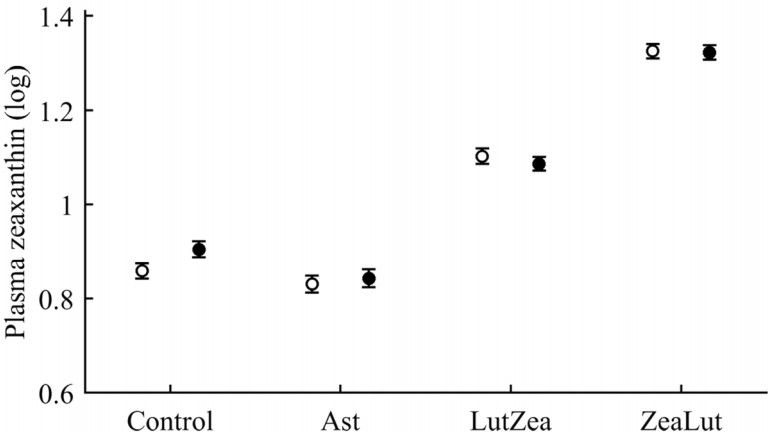

**Figure 8 Levels of lutein and zeaxanthin in plasma after diquat exposure depending on the carotenoid treatment.** Least square means ± SE were obtained from the models (see Methods and Table 3). Open circles: Control birds; Solid circles: Diquat-treated birds.

and 4). ZeaLut birds showed the highest tocopherol concentrations in the legs (both $P < 0.005$ when compared to LutZea and controls; $P = 0.085$ when compared to Ast).

### Plasma and internal tissues

With regard to circulating carotenoids, lutein showed a significant diquat × CAR interaction (Table 3). Among CAR groups, only controls showed significantly higher lutein levels with diquat ($P = 0.039$; control: 0.98 ± 0.01; diquat: 1.02 ± 0.01, log-values; Fig. 8). In the case of zeaxanthin, although the CAR × diquat interaction was non-significant ($P = 0.200$; Table 3), the post hoc comparison within the control-CAR group showed a similar diquat effect ($P = 0.033$; control: 0.86 ± 0.02; diquat: 0.91 ± 0.02; Fig. 8). No other pigment showed detectable levels in plasma.

With regard to plasma vitamins, tocopherol was unaffected by the diquat × CAR interaction (Table 3). Nonetheless, diquat showed a significant effect (Table 4), with tocopherol values decreasing after the exposure (control: 1.08 ± 0.02; diquat:

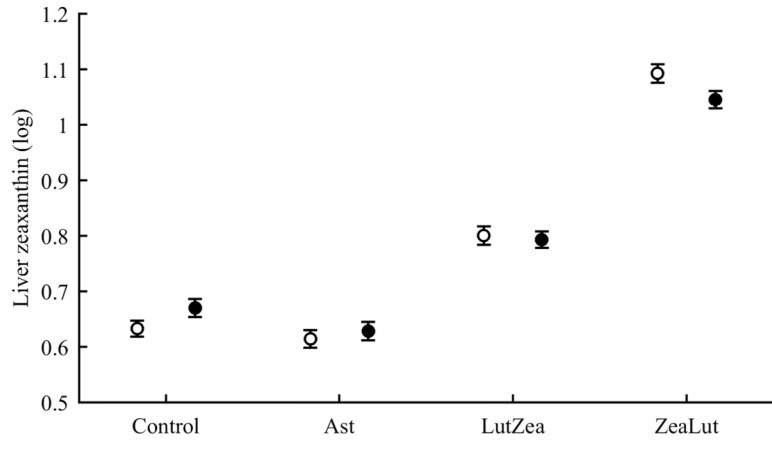

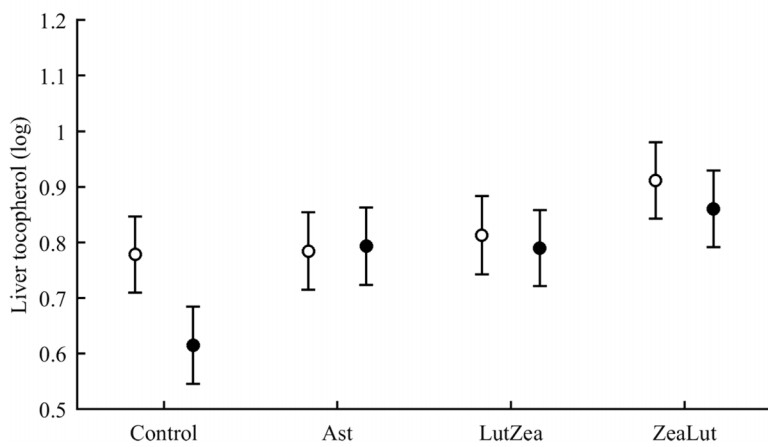

**Figure 9  Levels of zeaxanthin and tocopherol in the liver after diquat exposure depending on the carotenoid treatment.** Least square means ± SE were obtained from the models (see Methods and Table 3). Open circles: Control birds; Solid circles: Diquat-treated birds.

1.03 ± 0.02). No factor or interaction was significant in the case of plasma retinol (all $P > 0.10$; Table 4).

In the liver, the diquat × CAR interaction did not affect lutein levels (Table 3). The best-fitted model reported a strong significant CAR effect (Table 4). LutZea and ZeaLut birds did not differ ($P = 0.103$) and showed the highest lutein levels (Fig. S5). The other comparisons always reported $P < 0.001$, and the Ast group showed the lowest value (Table S3). In contrast, liver zeaxanthin showed a significant CAR × diquat interaction (Table 3). This effect was mostly due to diquat reducing zeaxanthin levels in ZeaLut birds ($P = 0.028$), and a trend in the opposite direction among controls ($P = 0.064$; Fig. 9). Importantly, such as in the case of astaxanthin in ornaments, the CAR factor ($P < 0.001$) reported increasing liver zeaxanthin values in the following order: Ast, control, LutZea and ZeaLut (all comparisons: $P < 0.040$).

With regard to liver vitamins, tocopherol was affected by CAR × diquat (Table 3). Among CAR groups, only control values showed a diquat effect on tocopherol, i.e. a decline

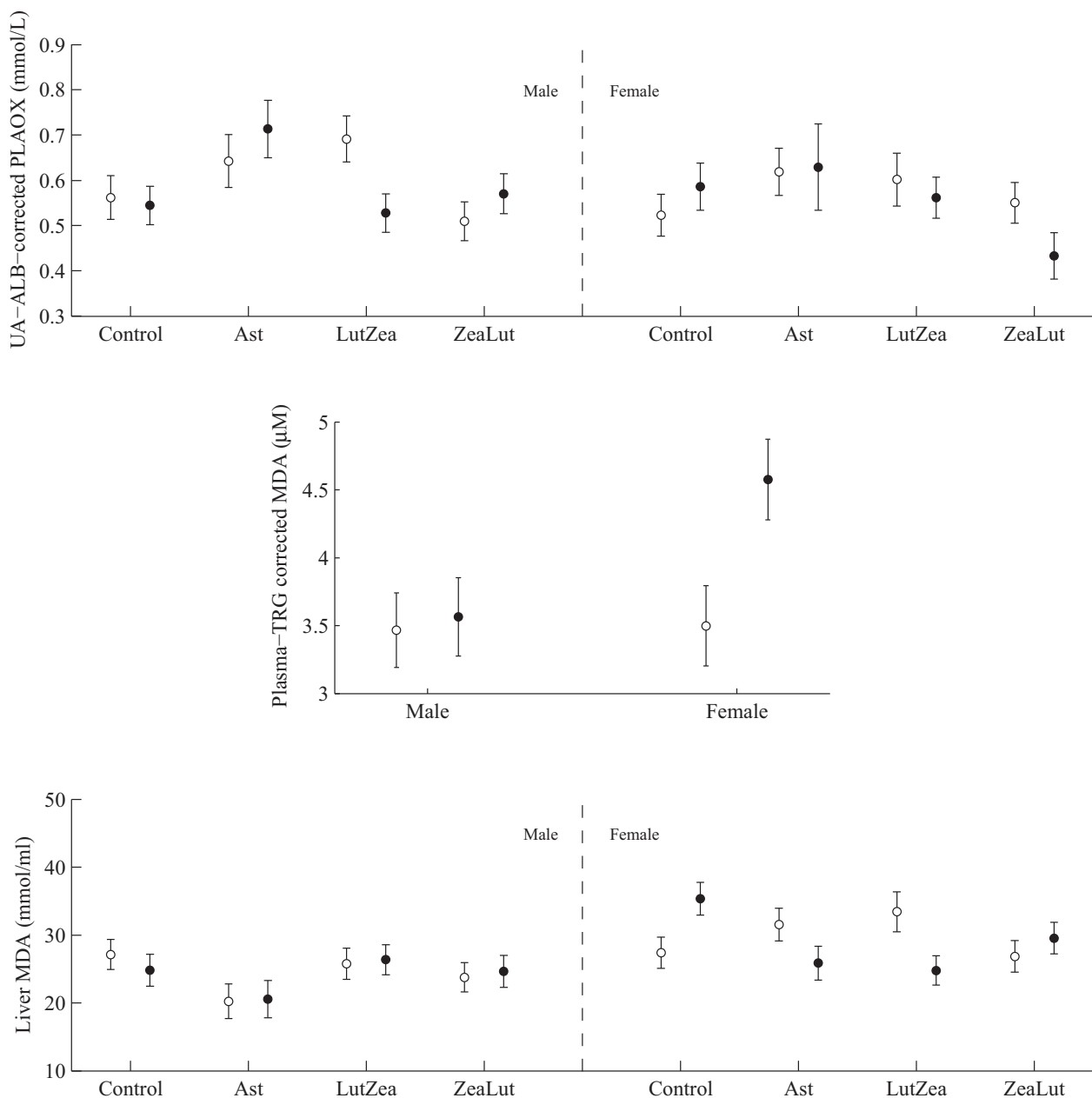

**Figure 10 Levels of oxidative stress biomarkers after diquat exposure depending on the carotenoid treatment.** Least square means ± SE from the models (see Methods and Table 3). Open circles: Control birds; Solid circles: Diquat-treated birds.

($P = 0.001$; Fig. 9). The CAR factor in the same model ($P < 0.001$) indicated significant differences following the order showed above for liver zeaxanthin (all $P < 0.003$), but here control and Ast birds did not differ ($P = 0.699$). In the case of liver retinol, both free and esterified retinol forms were detected, the two values being added for analyses (i.e. vitamin A). This variable was unaffected by CAR × diquat (Table 3) but showed a significant CAR effect (Table 4). LutZea and ZeaLut birds did not differ ($P = 0.133$), with Ast animals reporting the highest level, and control birds the lowest (other $P < 0.001$; Fig. S5).

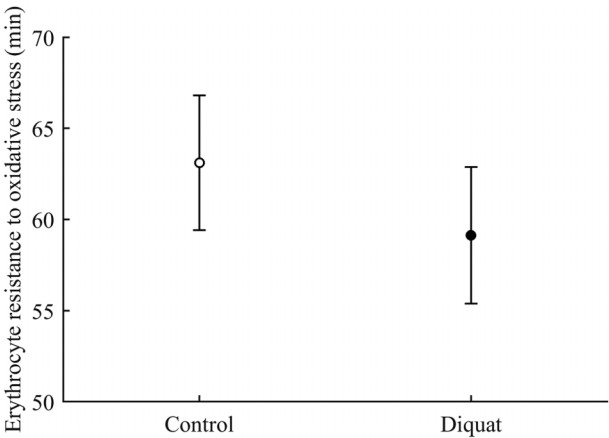

**Figure 11 Effect of the diquat treatment on the erythrocyte resistance to oxidative stress.** Least square means ± SE from the models (see Methods and Table 4). Open circles: Control birds; solid circles: diquat birds.

In the subcutaneous fat, no carotenoid or vitamin was affected by CAR × diquat (all $P$ values > 0.80; Table 3). The best-fitted models always reported a significant CAR effect (Table 4; Fig. S6; except for tocopherol). In the case of lutein, all groups differed from each other (all $P < 0.001$), except ZeaLut vs. control ($P = 0.915$). The LutZea group reported the highest lutein levels, and Ast birds the lowest. For zeaxanthin, LutZea birds tended to show higher values than controls ($P = 0.062$), with other groups significantly differing from each other (all $P < 0.012$). ZeaLut birds presented the highest zeaxanthin values, whereas Ast again showed the lowest. Tocopherol was not affected by any factor or interaction (all $P > 0.10$; Table 3; Fig. S7). With regard to retinol, all the groups differed from each other (CAR factor in Table 4), except ZeaLut and LutZea ($P = 0.955$). Ast and control birds showed the highest and lowest values, respectively (all $P < 0.001$; Fig. S6).

### Oxidative stress biomarkers

PLAOX showed a three-way CAR × diquat × sex interaction (Table 3; Fig. 10). Diquat decreased hydrosoluble antioxidant levels in LutZea males ($P = 0.02$), showing a trend in the same direction in females, but in the ZeaLut group ($P = 0.06$; Fig. 10). No factor or interaction remained (all $P > 0.18$) when removing uric acid and albumin covariates (though they showed $P < 0.057$).

In plasma MDA, CAR × diquat was non-significant ($P = 0.466$; Table 3), but diquat × sex interacted (Table 4; Fig. 10). Diquat-treated females showed higher lipid peroxidation than control females ($P = 0.001$; males did not differ: $P = 0.752$). The interaction did not change ($P = 0.008$) when removing the triglyceride covariate. The CAR group was never significant ($P > 0.5$). In liver MDA, the three-way interaction again arose (Table 3; Fig. 10). Diquat increased MDA values in control females ($P = 0.009$), but decreased MDA in LutZea ($P = 0.014$) and Ast (but at $P = 0.079$) females. Moreover, diquat control-CAR females also tended to endure higher liver MDA values than diquat ZeaLut females ($P = 0.068$). No difference was

found in males (all $P > 0.10$). The CAR group in the model was not significant ($P = 0.289$). No factor or interaction reported significant terms in heart MDA (Table 4; all $P > 0.12$).

Finally, in the case of erythrocyte resistance to oxidative stress, the CAR × diquat interaction only showed a weak trend toward significance ($P = 0.090$; Table 3), but the best-fitted model reported a significant diquat effect (Table 4; Fig. 11). The CAR factor was not significant (all $P > 0.50$).

## DISCUSSION

Our results as a whole suggest that the availability of certain carotenoids in the diet and the level of oxidative stress can coexist and interact to produce pigmentation in avian ornaments. As predicted, a higher dietary content of lutein vs. zeaxanthin (LutZea) led to a higher papilioerythrinone accumulation in the red ornaments, whereas the opposite (ZeaLut) led to a higher astaxanthin deposition. Birds fed with higher zeaxanthin and lutein proportions showed the reddest ornaments, but the first (ZeaLut) showed the reddest traits (eye rings and legs) at the end of the study. Furthermore, the oxidative challenge produced redder bills and higher astaxanthin deposition in the bare parts of some birds, the latter depending on tocopherol levels in the same tissue.

### Covariation between vitamins and carotenoids

The carotenoid treatments affected tocopherol levels in tissues. However, this only partially agreed with diet composition, which showed the highest vitamin values in the ZeaLut and, particularly, the Ast groups (Table 1). In legs, plasma and liver, only ZeaLut birds showed higher tocopherol levels than other groups. The lack of high tocopherol values in Ast partridges in these tissues could be explained by astaxanthin interfering with vitamin absorption (e.g. *Giraudeau et al., 2013*; also below). Nonetheless, both Ast and ZeaLut groups showed the highest tocopherol levels in the other ornaments. In the case of retinoids, Ast birds also showed the highest value in liver and fat, but ZeaLut and LutZea groups did not differ.

Results also suggest that carotenoids protected vitamin E from oxidative stress. In the bill, eye rings, and liver, diquat decreased tocopherol levels, but only among birds that did not receive carotenoid supplements. This suggests that a higher carotenoid availability among CAR-treated birds buffered tocopherol consumption due to free radicals, which supports the idea of mutual recycling and protective roles between tocopherol and carotenoids (*Mortensen, Skibsted & Truscott, 2001*; *Catoni, Peters & Schaefer, 2008*; *Surai, 2012*). The only exception was the diquat-mediated reduction in tocopherol levels in the bill of LutZea birds. Regardless, we must consider that, among carotenoids, lutein (i.e. the most abundant carotenoid in LutZea birds) is the weakest antioxidant (*Britton, 1995*; *Martínez et al., 2008*; see also below).

To discriminate carotenoid effects from the influence of vitamin variability in the diet, all the statistical models were controlled for tocopherol and retinoid levels in the tissues. The problem of collateral variation of antioxidants in a supplemented diet has mostly been ignored in experiments aiming to strictly manipulate dietary

carotenoid levels. For instance, *Stirnemann et al. (2009)*, *Toomey & McGraw (2011)* and *Toomey & McGraw (2012)* have used the same beadlets including zeaxanthin and tocopherol, but vitamin levels were not considered in their analyses. The addition of other antioxidants as excipients in carotenoid supplements would protect carotenoids during storage. In the other direction, the addition of carotenoids to the pelleted food could also have protected antioxidant vitamins in the basal diet, thus disrupting the original covariation among the levels of different compounds (Table 2; *Catoni, Peters & Schaefer, 2008*). In any event, we must note that diquat effects within a particular carotenoid treatment were independent of vitamin variability in that group as both diquat and control birds should have received the same vitamin amounts. In summary, results must be carefully interpreted in the light of vitamin covariation.

## Metabolic pathway of dietary carotenoids

We predicted that birds supplemented with astaxanthin should produce the most pigmented ornaments as biotransformation is not required. Surprisingly, dietary astaxanthin was apparently not absorbed. It was not detected in blood and other internal tissues. Moreover, astaxanthin seems to have interfered with lutein, zeaxanthin and tocopherol acquisition, as the circulating levels of these molecules declined in Ast birds. Consequently, ketocarotenoid deposition in ornaments and trait redness were reduced. Carotenoid competition during intestinal absorption and/or incorporation into the chylomicrons (e.g. *Tyssandier et al., 2002*; *Canene-Adams & Erdman, 2009*) can be argued considering the literature on humans (reviewed in *Furr & Clark (1997)* and *van den Berg (1999)*). In birds, competitive interactions of beta-carotene vs. lutein or zeaxanthin during intestinal absorption have also been reported for poultry diets (*Wang et al., 2010*). Interestingly, in the opposite direction, flamingoes (*Phoenicopterus ruber*) fed with lutein or zeaxanthin were unable to absorb these two pigments, but were instead able to assimilate astaxanthin, which is used as a precursor for the main carotenoid in their feathers (i.e. canthaxanthin; *Fox & McBeth, 1970*; *McGraw, 2006*). Our partridges also differ from European storks (*Ciconia ciconia*) naturally feeding on crayfish (*Procambarus clarkii*) containing high astaxanthin concentrations because they showed redder skin and higher astaxanthin concentrations in blood than controls (*Negro & Garrido-Fernández, 2000*). Phylogenetic differences may explain this. Astaxanthin is common in waterbirds feeding on fishes and aquatic invertebrates (an important astaxanthin source), but not among other avian species (*McGraw, 2006*). The red-legged partridge is a terrestrial granivorous gallinacean, and thus, astaxanthin is probably infrequent in their natural diet. For this reason, the capacity for assimilating astaxanthin may not have evolved. Nonetheless, other granivorous (but passerine) birds are able to absorb canthaxanthin (*McGraw & Hill, 2001*; *Hill, 2002*), another carotenoid described in aquatic organisms (*McGraw, 2006*).

On the other hand, our manipulation mostly supports the biotransformation pathway proposed for red-legged partridge carotenoids (i.e. *García-de Blas et al., 2014*); that is, lutein acting as the main papilioerythrinone precursor, with zeaxanthin acting as the

main astaxanthin substrate. Lutein and zeaxanthin levels rose in the blood, liver and fat according to their relative abundance in the diet. Similarly, papilioerythrinone and astaxanthin in ornaments increased in higher amounts in LutZea and ZeaLut groups, respectively. The results support previous correlative findings in the same species (*García-de Blas, Mateo & Alonso-Alvarez, 2015*) and demonstrate that the ketocarotenoids giving color to red-legged partridge ornaments are influenced by the availability of the most common hydroxycarotenoids in birds (*McGraw, 2006*). As previously mentioned, lutein and zeaxanthin are the most frequently described and abundant carotenoids in the food and blood of many bird species, as well as the most common substrates for red ketocarotenoids in ornaments, at least among non-aquatic species (*Surai et al., 2001*; *McGraw, 2006*). In passerines, lutein levels always prevail over zeaxanthin levels in both blood and diet, commonly at a 70:30 ratio (lutein:zeaxanthin) or higher (e.g. *McGraw et al., 2004*), which could also reflect the dietary content (*McGraw, 2006*). Our manipulation supports this for a gallinacean species. Moreover, *McGraw et al. (2004)* proposed that birds should prioritize zeaxanthin accumulation because this pigment would proportionally contribute more to coloring red ornaments compared to lutein. This has only been supported by correlations between the ratio of these two principal hydroxycarotenoids in the body and the ratio of pigments deposited in the ornaments (*McGraw & Gregory, 2004*; *García-de Blas, Mateo & Alonso-Alvarez, 2015*). Our experimental results also confirm this, and support, to some extent, the hypothesis that carotenoid-based signaling reveals an individual's capacity to find specific carotenoids in the environment (*Hill, 1994*; *Hill, 2002*).

Finally, the fact that astaxanthin and papilioerythrinone were only found in bare parts validates our previous findings (*García-de Blas, Mateo & Alonso-Alvarez, 2015*) and again supports the idea that biotransformation can take place in situ, at the colored trait, something only explored and described in passerines (*McGraw (2004)* and *McGraw (2009)* for eleven species; but see *del Val et al. (2009a)* and *McGraw & Toomey (2010)* for two other passerine species). The two recent studies describing a candidate oxygenase for carotenoid biotransformation have detected the enzyme in both the liver and feather follicles of canaries (*Serinus canaria*; *Lopes et al., 2016*), but only in the bare parts (bill and legs) of zebra finches (*Mundy et al., 2016*). These differences between only two passerine species would suggest a large diversity in evolutionary constraints and strategies among species.

## Dietary hydroxycarotenoids contributing to color

Lutein and zeaxanthin supplementation attenuated the color decline observed throughout the breeding season in red-legged partridges (*Alonso-Alvarez et al., 2008*). Consistently with the highest rate of astaxanthin deposition in the ornaments, the ZeaLut treatment produced the reddest birds at the end of the study. We must note that statistical analyses testing the CAR effect only (Table 2) did not include data from birds treated with diquat at the last sampling event, which reduced the sample size by half. When color was tested by controlling the diquat effect, differences between the ZeaLut and LutZea group arose (Tables 3 and 4; Fig. 5). The fact that ZeaLut birds were the reddest suggests that

individuals could try to obtain the highest zeaxanthin amounts in the diet to generate ornaments with the highest astaxanthin levels (also *García-de Blas, Mateo & Alonso-Alvarez, 2015*). This scenario may support the involvement of an allocation trade-off between signaling and self-maintenance functions (*Møller et al., 2000*) based on a hypothetically scarce resource (i.e. zeaxanthin). On the other hand, the presence of papilioerythrinone in the same ornaments is probably due to the abundance of lutein in the diet and the contribution of papilioerythrinone to color (*García-de Blas et al., 2013*; *García-de Blas et al., 2014*). However, astaxanthin is the most conjugated carotenoid, and hence, the reddest (and most abundant) pigment in red-legged partridge ornaments. Nonetheless, it has been shown that variability in papilioerythrinone levels in the red head traits can also contribute to explaining color variation, at least in a correlational sample of these birds (i.e. *García-de Blas et al., 2013*).

## Oxidative stress and carotenoids

Results support that diquat indeed increased oxidative stress in our birds, although the challenge was apparently mild because no effect on body mass or egg production was detected. Diquat generates superoxide and hydrogen peroxide radicals and has been previously used in the same dose and species, reporting effects on blood antioxidant machinery and lipid peroxidation (*Galvan & Alonso-Alvarez, 2009*; *Alonso-Alvarez & Galván, 2011*). In the present study, partridges treated with diquat showed weaker erythrocyte resistance to hemolysis when blood was exposed to another free radical source (AAPH). This measure has been associated with long-term (months or years) survival in other bird species (*Alonso-Alvarez et al., 2006*; *Bize et al., 2014*). Moreover, diquat-treated females, but not males, showed higher levels of oxidative damage in plasma lipids. Independently of diquat treatment, females allocated lower carotenoid and tocopherol (i.e. antioxidants) amounts to ornaments than males (sex factor at $P < 0.05$ in most models), suggesting a higher investment in other reproductive traits (e.g. egg yolk). Female birds could be more sensitive to oxidative damage during reproduction due to the costs associated with antioxidant allocation to eggs (e.g. *Williams, 2005*). Accordingly, female red-legged partridges producing eggs with higher hatching success (probably linked to antioxidant content; *McGraw, Adkins-Regan & Parker, 2005*) endured higher lipid peroxidation in erythrocytes (i.e. *Alonso-Alvarez et al., 2010*). Similarly, diquat-treated females, but not males, showed higher lipid peroxidation in the liver than controls, but only among birds that did not receive carotenoid supplements. In fact, LutZea and Ast females treated with diquat even showed a decline in liver MDA values compared to controls of the same group (Fig. 10). This may support the antioxidant role of xanthophylls involved in coloration, at least for females. This role has been questioned repeatedly, at least for avian species (*Hartley & Kennedy, 2004*; *Costantini & Møller, 2008*; *Isaksson & Andersson, 2008*; but see *Simons, Cohen & Verhulst, 2012*).

Finally, results from circulating hydrosoluble antioxidants (PLAOX) were less consistent, showing declines in response to diquat in some carotenoid groups only, and depending on the sex (Fig. 10). Moreover, independently of diquat effects, higher

PLAOX levels in Ast and LutZea birds of both sexes compared to controls were found (Fig. 3). In contrast, PLAOX did not increase in ZeaLut partridges. The antioxidant power of each pigment is linked to the number of conjugated double bonds: 13, 11 and 10 for astaxanthin, zeaxanthin, and lutein, respectively (*Britton, 1995*; *Britton, Liaaen-Jensen & Pfander, 2009*; *Martínez et al., 2008*). Therefore, an increase in PLAOX among ZeaLut birds was predictable. Nonetheless, we must consider that PLAOX mostly assesses the presence of hydro-, but not lipid-soluble antioxidants (*Miller et al., 1996*; *Cohen, Klasing & Ricklefs, 2007*). Thus, a higher PLAOX may also be due to a compensatory mobilization of other antioxidants (e.g. vitamin C) to fight off a challenge of some type (*Costantini, Metcalfe & Monaghan, 2010*). This view particularly agrees with the highest PLAOX values in Ast birds. These animals did not show astaxanthin in plasma and even experienced lower plasma lutein, zeaxanthin and tocopherol levels than controls (above). Similarly, Ast birds did not show astaxanthin in the liver, but accumulated large amounts of vitamin A in this organ, perhaps to protect the liver from some toxic insult (*García-de Blas, Mateo & Alonso-Alvarez, 2015*). Anyway, we found only one study supporting this toxic effect, in which rats fed with astaxanthin endured an impairment of the liver enzymes involved in detoxification (*Ohno et al., 2011*). In summary, if PLAOX did not exclusively reveal the antioxidant capacity of circulating carotenoids, the lack of higher PLAOX values in ZeaLut birds could merely be due to other (hydrosoluble) antioxidants being not mobilized. Here the conclusion is that the antioxidant role of carotenoid cannot easily be addressed by PLAOX measures only.

## Oxidative stress and carotenoid biotransformation

Although the proximate cost of ketocarotenoid-based signaling in red-legged birds may, at least partially, involve increased foraging effort to obtain large zeaxanthin amounts (above), the requirement of biotransformation to produce red traits provides another substrate for natural selection. Birds exposed to diquat generated redder bills, which contradicts the constraining impact of oxidative stress on health (e.g. *Monaghan, Metcalfe & Torres, 2009*; *Dowling & Simmons, 2009*; *Costantini, 2014*). The results may, instead, support some response (perhaps hormetic; e.g. *Costantini, Metcalfe & Monaghan, 2010*) against a mild stressor, at least in terms of color expression, although the exact mechanism can only be deduced (see below).

We must anyway mention that, in contrast to our results, red-legged partridges exposed to the same diquat dose and duration in another experiment, but during the first weeks of life, produced paler red colors in adulthood (*Alonso-Alvarez & Galván, 2011*). We must nonetheless consider that adverse conditions during early periods of life are particularly damaging (*Metcalfe & Monaghan, 2001*). Young individuals may not have fully developed antioxidant machinery (*Metcalfe & Alonso-Alvarez, 2010*) to properly manage such an oxidative challenge. Here, pigment levels in partridge ornaments support the color findings. Carotenoid concentrations increased under diquat exposure. Interestingly, the increase in these tissues was detected for astaxanthin, but not papilioerythrinone.

We can provide two alternative or complementary proximate mechanisms to explain these findings. First, we may suggest that a large availability of superoxide and hydrogen peroxide (free radicals) derived from diquat redox cycling (*Koch & Hill, in press*) could favor those conditions required for oxygen addition to hydroxycarotenoids by the enzyme (i.e. more than dehydrogenation), and hence, a higher astaxanthin production. We must here remember that astaxanthin production from its substrate requires two oxygenation reactions, whereas papilioerythrinone would require one oxygenation but also a dehydrogenation (*McGraw, 2006*; *LaFountain, Frank & Prum, 2013*; *García-de Blas et al., 2014*). We must also consider that the hypothesized oxygenase (above) should require oxygen, as well as $Fe^{2+}$ cation and nicotinamide adenine dinucleotide phosphate (NADPH; a reducing agent; *Fraser, Miura & Misawa, 1997*; *Schoefs et al., 2001*). The second possibility would be that superoxide and hydrogen peroxide levels increased by diquat could act as redox signals (e.g. *Hurd & Murphy, 2009*) promoting oxygenase (but not dehydrogenase) transcription as a defensive or hormetic mechanism that would ultimately lead to carotenoid biotransformation. The two recent and simultaneously published articles describing the candidate oxygenase for converting yellow to red carotenoids in birds (*Lopes et al., 2016*; *Mundy et al., 2016*) show that the enzyme (i.e CYP2J19) is part of the well-known P450 cytochrome, which is involved in many detoxification reactions. Moreover, diquat has been shown to stimulate the transcription of similar oxygenase enzymes (i.e. heme-oxygenases) via redox signaling (i.e. via the Nuclear factor (erythroid-derived 2)-like 2 (Nrf2); e.g. *Black et al., 2008*; *Sun et al., 2011*; *Wilmes et al., 2011*). We may here argue that high superoxide or hydrogen peroxide radicals produced by natural processes (e.g. exercise, flying effort; *Costantini, Mirzai & Metcalfe, 2012*; *Jenni-Eiermann et al., 2014*) could activate a similar redox mechanism favoring oxygenase activity, which could explain why wild partridges are redder compared to captive birds whose flying capacity is restrained (*García-de Blas, Mateo & Alonso-Alvarez, 2015*).

Biotransformation due to oxidative stress, however, seems to be higher among birds with the highest availability of the main ketocarotenoid precursor; that is, ZeaLut birds (see in the eye ring; though $P = 0.057$; Fig. 6). This again supports the importance of acquiring enough quantity of specific carotenoids with the diet in a sexual signaling context (i.e. *Hill, 1994*; but see *Hill, 2011*). Nevertheless, the clearest effect was found in diquat-treated birds that did not receive any carotenoid supplementation (Fig. 6). The effect in these two CAR groups would agree with bill color findings (Fig. S2), although the interaction was non-significant. The diquat effect on non-supplemented birds could be due to better zeaxanthin availability in the blood (Fig. 8) and liver (Fig. 9) in this group. Higher circulating levels of zeaxanthin could be a consequence of an active mobilization from stores (liver) and/or better intestinal absorption, both for combating oxidative stress (e.g. *Alonso-Alvarez et al., 2008*; *McClean et al., 2011*; but see *Isaksson & Andersson, 2008*). Recent works suggest that xanthophyll absorption in the intestinal mucosa can be actively regulated by specific protein scavenger receptors such as the class B member 1 (SR-B1; *Hill & Johnson, 2012*; *Sato et al., 2012*). How diquat may have favored such receptors can only be speculated, but we could again

consider its potential influence on different redox signaling pathways (above; also e.g. *Cristóvão et al., 2009*; *Koch & Hill, in press*).

In any event, in order to test whether higher astaxanthin levels in ornaments are due to higher zeaxanthin availability (mobilization) in the body (i.e., not to higher biotransformation rates), we also added plasma or liver zeaxanthin levels as covariates in models testing bill and eye ring astaxanthin concentrations. As expected, a positive link between ornament astaxanthin and plasma zeaxanthin values was observed (also *García-de Blas, Mateo & Alonso-Alvarez, 2015*), but this did not change the CAR × diquat interaction or post hoc tests (always $P < 0.05$). Moreover, diquat did not increase zeaxanthin values in internal tissues in the other group showing increased astaxanthin deposition in ornaments (ZeaLut; Fig. 8).

Other results may still support the availability of carotenoid precursors as a key factor favoring biotransformation. Diquat decreased tocopherol values in eye rings and bills among birds that did not receive supplemented carotenoids in food (Fig. 7). When tocopherol levels in these bare parts are not statistically controlled for as a covariate, differences in astaxanthin levels among the same control birds (Fig. 6) disappear (both traits: $P > 0.60$), but in the eye rings of ZeaLut birds they become significant ($P = 0.036$). In other words, ZeaLut birds showed the highest astaxanthin levels in eye rings when exposed to diquat. This suggests that biotransformation can be even more stimulated by oxidative stress when the level of carotenoid precursors in the diet surpasses some threshold. When this is not the case, color is not impaired but tocopherol levels are probably consumed to control the challenge.

In summary, the overall results suggest that specific carotenoid precursors must be sufficiently available and that oxidative status must be well-adjusted in order to produce the most pigmented red ornaments. In agreement with this, redder integuments have also been observed in red-legged partridges exposed to other chemicals (i.e. pesticides and heavy metals) that induce oxidative stress (*Lopez-Antia et al., 2015a*; *Lopez-Antia et al., 2015b*; *Vallverdú-Coll et al., 2015*) or in zebra finches enduring experimentally reduced antioxidant (glutathione) levels (*Romero-Haro & Alonso-Alvarez, 2015*). The findings support the view that oxidative stress is not only a constraint for the expression of optimal phenotypes, but that mild levels are involved in many functions (*Jones, 2006*; *Metcalfe & Alonso-Alvarez, 2010*; *Isaksson, Sheldon & Uller, 2011*). Furthermore, the study supports claims from *Hill & Johnson (2012)* and *Johnson & Hill (2013)* hypothesizing that carotenoid-based traits could be signaling an individual's efficiency to manage oxidative stress. The results also validate the *Völker's (1957)* ideas suggesting that a good oxidative metabolism is necessary to biotransform carotenoids used in red coloration, which could explain why birds whose flying capacity was restrained by captivity became paler. However, in contrast to the works of Hill & Johnson (i.e., *Hill & Johnson, 2012*; *Johnson & Hill, 2013*), our experiment also suggests the parallel involvement of a resource allocation trade-off because the body levels of substrate carotenoids influenced coloration and even the impact of oxidative stress on biotransformation. In eye rings, under oxidative stress exposure, birds receiving the highest zeaxanthin levels in the diet were also those producing the highest amounts of the main ketocarotenoid (astaxanthin). Finally, we cannot conclude this discussion without applying

a life-history perspective. High levels of sexual signaling under high oxidative stress could constitute a sort of terminal investment, with individuals increasing their chances of reproducing when their perception of future survival becomes negative (*Velando, Drummond & Torres, 2006*; *Romero-Haro & Alonso-Alvarez, 2015*).

## ACKNOWLEDGEMENTS

We thank Ester Ferrero and Laura Ramirez for their help with the laboratory work and photographic analyses and Álvaro Galán for his help with the figures. We also thank Lorenzo Pérez-Rodríguez, Ana Angela Romero-Haro, Núria Vallverdú and Jaime Rodríguez their help during blood sampling and to Xurxo Piñeiro, Emiliano Sobrino, Fernando Dueñas and Luis Montó for partridge maintenance and Sarah Young for reviewing the English writing. We especially thank Prof Geoffrey Hill for his constructive review of the text.

### Funding

Esther García-de Blas was supported by a predoctoral grant (JAE-PRE) from the Consejo Superior de Investigaciones Científicas (CSIC) co-financed by Fondo Social Europeo (EU). This study was funded by Consejería de Educación y Ciencia, Junta de Comunidades de Castilla la Mancha (project ref.: PII1I09-0271-5037) and Ministerio de Economía y Competitividad (CGL2009-10883-C02-02 and CGL2015-69338-C2-2-P) from the Spanish Government. The funders had no role in study design, data collection and analysis, decision to publish, or preparation of the manuscript.

### Grant Disclosures

The following grant information was disclosed by the authors:
Consejo Superior de Investigaciones Científicas: JAE-PRE.
Consejería de Educación y Ciencia, Junta de Comunidades de Castilla la Mancha: PII1I09-0271-5037.
Ministerio de Economía y Competitividad: CGL2009-10883-C02-02 and CGL2015-69338-C2-2-P.

### Competing Interests

The authors declare that they have no competing interests.

### Author Contributions

- Esther García-de Blas performed the experiments, wrote the paper, prepared figures and/or tables, reviewed drafts of the paper.
- Rafael Mateo performed the experiments, contributed reagents/materials/analysis tools, reviewed drafts of the paper.
- Carlos Alonso-Alvarez conceived and designed the experiments, performed the experiments, analyzed the data, contributed reagents/materials/analysis tools, wrote the paper, prepared figures and/or tables, reviewed drafts of the paper.

## Animal Ethics

The following information was supplied relating to ethical approvals (i.e., approving body and any reference numbers):

The experimental protocol was approved by the University of Castilla-La Mancha's Committee on Ethics and Animal Experimentation (reference number 1011.01).

## Data Deposition

The raw data has been supplied as Supplemental Dataset Files.

## Supplemental Information

Supplemental information for this article can be found online at http://dx.doi.org/10.7717/peerj.2237#supplemental-information.

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
