# Peer review of "Specific carotenoid pigments in the diet and a bit of oxidative stress in the recipe for producing red carotenoid-based signals"

_PeerJ, doi:10.7717/peerj.2237_

## Round 0.1 · original submission · Minor Revisions

I apologize for the delay in sending you this decision. We were waiting for another review, which did not happen. I have read your manuscript and the attached review and I agree with this reviewer that minor revisions are necessary.

·

Basic reporting

This is a solid study. In my "comments to Authors" I give many small suggestions for improvement but I find no serious flaw in the study.

Experimental design

The experimental design seems solid. More justification of use of the specific diaquat treatment would strengthen the paper, but I think the treatment can be justified.

Validity of the findings

The patterns revealed by this study are certainly valid and important. I do not agree with all interpretations, but the viewpoints of the authors are valid and important.

Additional comments

Carotenoid-based ornaments are honest signals of individual quality. This has been well established through many studies in the late twentieth and early 21st centuries. What remains uncertain is how this association between pigmentation and individual quality comes about.
The Alonso-Alvarez lab has been at the forefront of designing experiments to test alternative hypotheses for how carotenoid coloration and particularly the concentration of ketolated carotenoids serves as an honest signal of individual condition. The study submitted here is a valuable contribution to the effort to understand the signal content of carotenoid coloration.
I have some questions about protocols and methods and I have some concerns (and certainly differences of opinion) about how data are interpreted, but my overall assessment is that this is a solid contribution to the field.

Abstract lines 20-23 – the logic of how the results suggest a trade-off is not clear. I would either more explicitly explain why the differences caused by supplementation indicate a trade-off or omit this speculation from the abstract. This study includes an array of measurements and manipulations that sets up an unprecedented and extremely valuable opportunity to evaluate interactions among physiological variables that may be related to color expression. I suggest reframing some aspects of the abstract—and perhaps discussion—to emphasize the study’s overall focus on complex interactions.

Page 4 Lines 5-6 – With respect to Simons et al. 2012, I don’t think the evidence of carotenoids as immune stimulants is nearly conclusive; note that only response to PHA injection was significantly related to either carotenoids or trait redness, among 5 measurements of immune response (Fig. 2). While many studies find some correlative support for links between some aspects of immune response with some aspects of carotenoid prevalence or coloration, I think it is important to keep a broader and more objective perspective on the state of the debate of carotenoids as immune boosters (or not).

Page 7 Lines 13-18 – This seems like a core justification of your experimental design and interpretation. As such, I would devote more space to explaining how this works. How, specifically—mechanistically—might an increased production of RONS help ketolase activity? Also, might RONS directly modify carotenoids—and if so, how?

Page 8 Lines 12-14 – It’s important to further justify / describe the quantities of carotenoids the partridges ingested from the various diets and how the levels provided relate to what a wild partridge would ingest. Is the quantity of carotenoids similar to what we might expect to be “natural” for this species? The authors focus on ratios of pigment types, but if the overall availability of carotenoids is well above or below what a wild partridge would have access to, then that completely changes how the results will be interpreted. The poultry industry often gluts the diets of food or egg birds to adjust the coloration of products for consumption, and so the numbers from poultry research may not have biological relevance. The quantities of carotenoids consumed and therefore physiologically available is key to this study. The authors should cite the following in considering carotenoid dose:
Rebecca E. Koch, Alan E. Wilson, and Geoffrey E. Hill. 2016 The Importance of Carotenoid Dose in Supplementation Studies with Songbirds. Physiological and Biochemical Zoology 89:1, 61-71.

Page 8 Lines 27-29 – Can’t high temp and pressure also turn carotenoids into pro-oxidants? Do you have any verification that the nature of the carotenoids themselves wasn’t altered by the pelleting process?

Page 8 Line 30 – If tocopherol is going to be important to your results / discussion, I think it needs more introduction. What is it, and what role does it play in the interactions you are studying?

Page 10 Lines 10-12 – It is important to take the time to justify the diquat dose. What symptoms does it induce in the birds? Has it been shown to induce oxidative damage? In justifying the use of diquat and considering the importance of correct dose the authors should cite:
Koch, R. E. and Hill, G. E. (2016), An assessment of techniques to manipulate oxidative stress in animals. Funct Ecol. Accepted Author Manuscript. doi:10.1111/1365-2435.12664


Page 10 Line 18 – I question the quantification of the spectral data. The focus of this paper is on carotenoid ketolation. High versus low performance regarding carotenoid ketolation is typically manifest in the ratio of yellow unmodified pigments to red modified pigments in tissue, which affects the position of the reflectance curve, which is measured as hue. If the hue measure presented by the authors (and then confounded with brightness) is a measure of the position of the reflectance peak, then the authors should explain this more clearly. I’m not sure why the authors followed the techniques of Saks et al., which was used to measure yellow coloration in Greenfinches, when Montgomerie has detailed methods for red coloration in his 2006 book chapter:
Montgomerie R (2006) Analyzing colors. In: Hill & McGraw, editor. Bird Coloration: Volume 1, Mechanisms and Measurements. Cambridge, Massachusetts: Harvard University Press. pp. 90–147.


Page 20: I interpreted, from the methods, that diquat was administered after “time 2”—day 48. Is that correct? If so, how are the coloration results shown here—from day 48—relevant to diquat effects? More importantly, it seems that the “control” coloration values are missing from the graphs in Fig. 5.

Page 22, lines 2-7: Wouldn’t these results (effects of supplementation on pap. concentration) be more appropriately presented in section 3.2?

Page 21-22: Why are the effects of diquat on lutein/zeaxanthin concentrations not described here? Were they not considered important—and if so, why not?

Page 22, lines 8-14: By this point in the paper, I think the reader may have forgotten the purpose of including tocopherol. A one sentence statement of the relevance of tocohperol would be very useful.

Page 22, lines 9-14: I’m guessing you report values only for lutein and zeaxanthin because the other pigments are not found in circulation. If so, that should be stated explicitly in this part of the paper.

Page 29, lines 12-14: I’m not sure the logic here is clear. What is the interaction between carotenoids and tocopherol?

Page 30, lines 19 to 27. To complete this discussion of constraints on absorption of ketolasted carotenoids, the authors should note that House Finches with red feather coloration readily take up canthaxanthin:
Hill 2002. Red bird in brown bag. Oxford Press.

Even more interesting, American Goldfinches, which never use ketolated carotenoids as feather pigments, absorbed canthaxanthin from diet and grew red feathers.
McGraw, K. J. and G. E. Hill. 2001. Carotenoid access and intraspecific variation in plumage pigmentation in male American Goldfinches (Carduelis tristis) and Northern Cardinals (Cardinalis cardinalis). Funct. Ecol. 15:732-739.


Page 31, lines 19-20: I just looked through Endler 1980 and he never speculates about foraging for specific carotenoids. He only talks about “good diets” and “poor diets”. So far as I know, the first time foraging for specific carotenoids was discussed is in:

Hill, G. E. 1994. Trait elaboration via adaptive mate choice: sexual conflict in the evolution of signals of male quality. Ethology, Ecology and Evolution 6: 351-370.
And discussed in more detail in
Hill, G. E. 2002. A red bird in a brown bag. Oxford Univ. Press, Oxford.

Page 32, lines 18-19: I think the documented effects of diquat need to directly stated somewhere near the beginning of the results. Much of the inference drawn from the paper depends on diquat increasing free radical production in tissues relevant to the research questions without unwanted targeted effects such as changing uptake of carotenoids in the gut. The more solidly the authors can justify their use of diquat, the stronger the paper becomes.

Page 36, lines 6-8: I don’t understand the logic that supports a trade-off. Trade-off between what, and what?

METHODS/RESULTS: I suggest consistently using either “time 1/2/3” or “day 0/48/82” across methods and results. Shifting between designations left me confused about what was performed when.

RESULTS: I would further clarify that that section 3.2 deals with the effects of supplementation only over time, while 3.3 looks at the effects of diquat treatment (on one time point only). Although this is mentioned in the statistical methods, it would be effective to briefly reiterate these distinctions—otherwise, discussions of the effects of supplementation on coloration seem repetitive.

RESULTS: I recommend restructuring the results section. I think it maybe could benefit from a cleaner separation of the results into “Coloration,” “Carotenoid composition,” and “Oxidative stress” subheadings, discussing within each subsection which treatments had significant effects (or not). As currently written, the results section is more confusing than necessary because it often shifts among time points, treatments, ornaments, or other physiological parameters, and the reasons why some (but not all) measurements are mentioned in any subsection are not clear. I think that if you instead divide the section into three subsections that describe the effects of diquat and supplementation on each of the three main physiological systems, it will be much clearer.

RESULTS 3.2: If you measured ornament carotenoid pigment composition at the end of the experiment, why don’t you present the effects of supplementation treatments on pigments (in non-diquat birds) here? I see the results on carotenoid pigments in section 3.3 (on diquat), but it seems like it would be applicable here in section 3.2 as well.


RESULTS and DISCUSSION: The interaction between diquat treatment and sex is a particularly interesting and unexpected finding of this study, and should be further highlighted. Why might we expect these differences? Could there be sex-based differences in strategy, and allocation to ornamentation, reproduction, and antioxidants?

DISCUSSION: The concept of hormesis should be further emphasized in the discussion, because it may be critical to understanding how diquat could increase coloration of ornaments.

---

## Round 0.2 · accepted · Accept

Thank-you for carefully addressing the review comments.